# MEDINSIGHTBENCH: EVALUATING MEDICAL ANALYTICS AGENTS THROUGH MULTI-STEP INSIGHT DISCOVERY IN MULTI-MODAL MEDICAL DATA

## ABSTRACT

In medical data analysis, extracting deep insights from complex, multi-modal datasets is essential for improving patient care, increasing diagnostic accuracy, and optimizing healthcare operations. However, there is currently a lack of high-quality datasets specifically designed to evaluate the ability of large multi-modal models (LMMs) to discover medical insights. In this paper, we introduce MedInsightBench, the first benchmark that comprises 332 carefully curated medical cases from cancer genomics atlas data, each annotated with thoughtfully designed insights. This benchmark is intended to evaluate the ability of LMMs and agent frameworks to analyze multi-modal medical image data, including posing relevant questions, interpreting complex findings, and synthesizing actionable insights and recommendations. Our analysis indicates that existing LMMs exhibit limited performance on MedInsightBench, which is primarily attributed to their challenges in extracting multi-step, deep insights and the absence of medical expertise. Therefore, we propose MedInsightAgent, an automated agent framework for medical data analysis, composed of three modules: Visual Root Finder, Analytical Insight Agent, and Follow-up Question Composer. Experiments on MedInsight-Bench highlight pervasive challenges and demonstrate that MedInsightAgent can improve the performance of general LMMs in medical data insight discovery.

## 1 INTRODUCTION

Recent advancements in medical data analysis using large multi-modal models (LMMs) have significantly improved clinical diagnosis (Mendoza et al.; Sun et al., 2025a; Xu et al., 2024). Medical insight detection that transforms heterogeneous data (e.g., pathological images) into actionable insights is crucial to improve diagnostic accuracy, guide treatment decisions, and enable new scientific discoveries (Zhan et al., 2025; Lu et al., 2024).

Despite recent strides in LMMs for combined visual–language reasoning (Mendoza et al.; Sun et al., 2025a; Xu et al., 2024), their diagnostic accuracy and medical insight detection in real-world clinical settings remains limited (Fan et al., 2025; Schmidgall et al., 2024). Existing benchmarks mainly probe surface-level competencies, such as retrieving overt facts or answering direct questions (Pandit et al., 2025; Shang et al., 2025). They overlook higher-order clinical cognition, which includes uncovering occult pathological relationships, formulating pathophysiologically grounded hypotheses, and integrating multi-modal evidence for prognostic inference (Wu et al., 2025; Tang et al., 2025). Therefore, there is a requirement for benchmarks that can assess whether LMMs can automatically discover, synthesize, and generate reliable, clinically meaningful insights from pathology data.

To facilitate a comprehensive evaluation of insight discovery in pathology, we propose MedInsight-Bench, a novel benchmark that includes high-quality medical images, explicit analytical goals to guide exploration, and question-insight pairs. MedInsightBench comprises 332 cases and 3,933 insights across six categories, utilizing a raw dataset from public cancer pathology resources. Our methodology involves downsampling WSI files to PNGs, segmenting report text into related evidence snippets with human verification, and deriving concise analysis goals from the logical relationships among the generated questions. Based on MedInsightBench, we conducted a comprehensive evaluation of five LMMs, assessing their effectiveness in insight discovery within pathology.

The evaluation of LMMs, such as GPT-4o (OpenAI, 2024) and Deepseek-VL2 (Wu et al., 2024), on MedInsightAgent reveals significant limitations. LMMs often struggle with multi-step analytical workflows that require image parsing, statistical reasoning, domain-constrained inference, and verifiability. Furthermore, LMMs show limited domain expertise, unstable chain-of-thought reasoning, and poor interpretability, all of which impact insight reliability and clinical utility. To address these issues, we propose MedInsightAgent, a multi-agent collaborative framework that has three main components: (i) Visual Root Finder extracts key image cues and background knowledge to generate initial root questions; (ii) Analytical Insight Agent analyzes image regions for each question to produce grounded answers and insights; and (iii) Follow-Up Question Composer generates iterative, derivative questions to enable deeper and more exploratory discovery. Agents exchange constrained information and iterate to produce deeper, more reliable, and more interpretable insights.

In our experiments, we benchmark multiple LMMs and agent frameworks on MedInsightBench using a comprehensive evaluation protocol, including Insight Recall, Precision, $F1$, and Novelty. The results highlight key challenges in automated medical insight discovery and show that our MedInsightAgent significantly enhances the insight discovery performance of base LMMs. In summary, our contributions are as follows.

- We introduce a novel multi-modal benchmark for the discovery of medical insight. The data set pairs pathology images with text and includes hierarchical tasks and validated metrics to assess the quality of knowledge.
- We design a multi-agent collaborative framework for insight discovery. The framework formalizes agent roles and interaction protocols to combine local visual analysis, cross-sample inference, and domain knowledge.
- We provide extensive empirical analysis on several baseline LMMs and on our multi-agent system. The experiments show the discriminative power of the benchmark and demonstrate that the multi-agent approach improves the precision and interpretability of the information.

## 2 RELATED WORKS

### 2.1 MEDICAL DATA ANALYSIS

Recent research has introduced benchmarks and frameworks for evaluating large language models (LLMs) and agent systems on medical reasoning and data analysis tasks. Several datasets emphasize multi-step clinical reasoning and multi-modal expert questions, including MedAgentsBench (Tang et al., 2025), MedAgentBench (Jiang et al., 2025), MedCaseReasoning (Wu et al., 2025), MedXpertQA (Zuo et al., 2025), and the Chinese CMB (Wang et al., 2024). Other work addresses interactive clinical workflows and multi-agent collaboration. AI Hospital (Fan et al., 2025), Agent-Clinic (Schmidgall et al., 2024), 3MDBench (Sviridov et al., 2025), and MedChain (Liu et al., 2024) simulate multiturn patient-clinician interactions, while MMedAgent-RL (Xia et al., 2025), MedAgentBoard (Zhu et al., 2025), and the MAD framework (Smit et al., 2023) explore multi-agent training and collaboration strategies, finding that multi-agents do not always outperform strong LMM. Finally, some studies address evaluation gaps and modality-specific challenges, such as Med-Hallu (Pandit et al., 2025), which focuses on hallucination detection, and work that disentangles knowledge from reasoning to expose benchmark inflation. MedRepBench (Shang et al., 2025) evaluates vision-language models to interpret complex medical reports, and Med3DInsight (Chen et al., 2025) improves 3D image understanding by leveraging 2D LMM pretraining. In contrast, our work centers on multi-step and in-depth explorative insight discovery in the medical domain, pushing the boundaries of traditional evaluations to uncover deeper, more nuanced insights.

### 2.2 DATA INSIGHT AGENTS AND BENCHMARKS

Some work on LLM-driven data analysis has produced benchmarks, datasets, and agentic frameworks that move beyond single-query answers to multi-step analytical workflows. Text-to-SQL efforts include FinSQL (Zhang et al., 2024), Spider 2.0 (Lei et al., 2024), EHRSQL (Lee et al., 2022), and PRACTIQ (Dong et al., 2025), which address domain-specific querying, complex multi-step SQL, and conversational ambiguity. For visualization, VisEval (Chen et al., 2024a) offers a large evaluation system, while MatPlotAgent (Yang et al., 2024) and nvAgent (Ouyang et al., 2025) propose multi-agent workflows to iteratively generate and validate visualizations, showing signif-

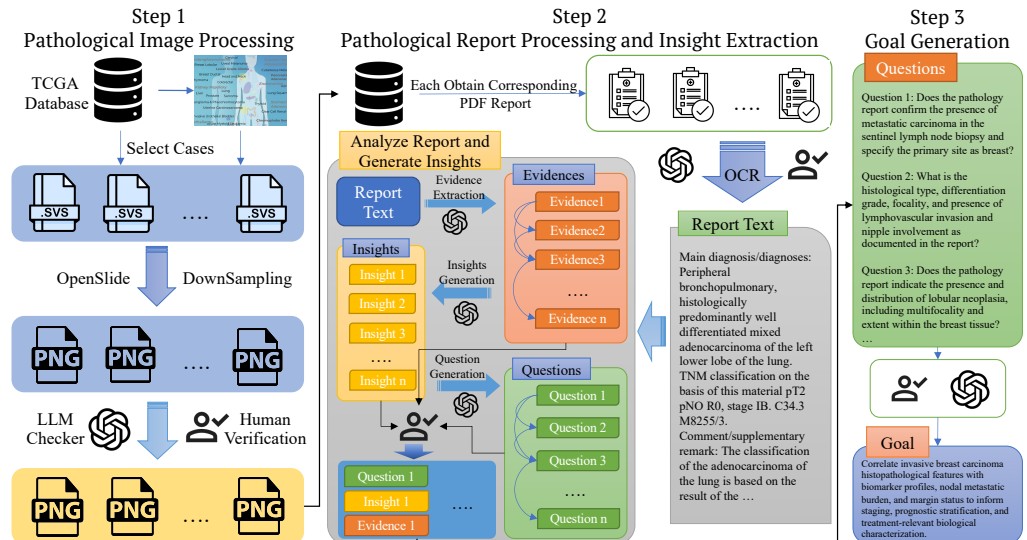

Figure 1: The dataset construction pipeline of MedInsightBench. This pipeline consists of 3 steps: 1) **Pathological Image Processing**. WSIs are standardized and quality-checked. 2) **Report Processing & Insight Extraction**. Reports are converted to text, insights and questions are generated, and verified by experts. 3) **Goal Generation**. An overarching analysis goal is synthesized from the questions and validated for guiding the analysis.

icant improvements. In addition, InfiAgent-DABench (Hu et al., 2024) introduces a broad benchmark for assessing LLM-based data analysis agents, and DAgent (Xu et al., 2025) extends this by generating complete analytical reports from relational databases. Other works target end-to-end insight generation. For example, InsightBench (Sahu et al., 2025), InsightPilot (Ma et al., 2023), InsightLens (Weng et al., 2025), and an LLM-based SQL decomposition approach (Pérez et al., 2025) cover multi-step discovery, autonomous exploration, and insight organization. Our MedInsightBench is the first comprehensive and high-quality benchmark for medical insight discovery.

# 3 MEDICAL INSIGHT BENCHMARK

## 3.1 PRELIMINARY STUDY OF INSIGHT DISCOVERY TASK

In the insight discovery task, there are a variety of viable approaches. For tabular data insights, multi-agent pipelines have proven particularly effective. Two state-of-the-art multi-agent paradigms are Pérez et al. (Pérez et al., 2025), which uses SQL to extract information from structured tables, and Agent Poirot (Sahu et al., 2025), which relies on Python scripts and standard data-analysis libraries to retrieve the relevant statistics and evidence.

For insight discovery in image-modal data, we considered both LMM and agentic framework paradigms. Given an image and a pre-specified goal, LMM can directly produce a set of analytical insights after reasoning. By contrast, a multi-agent pipeline proceeds in multiple steps: it generates a sequence of goal-directed questions, answers those questions, and finally synthesizes insights.

Guided by InsightBench (Sahu et al., 2025), we identified several design requirements for a high-quality medical image-based insight benchmark:

1. **Medical image quality and completeness**: The images must clearly and fully depict the target content so that relevant features are observable.
2. **Explicit analytical goal**: The analysis goal must be unambiguous and state the intended focus, such as relevant comparison metrics, axes, or dimensions of analysis, etc.
3. **Question–insight consistency**: Each insight must be supported by a clear, well-formed question and grounded in solid evidence. The questions should be comprehensive and multidimensional, while the insights should be meaningful, informative, and well-rounded.

Based on these principles, we constructed a novel medical insight discovery benchmark. The dataset construction pipeline is described in detail in the next section.

## 3.2 DATA CONSTRUCTION

From our preliminary study, we identified the key priorities and objectives for dataset construction: (i) high-quality and comprehensive medical images; (ii) an explicit and well-specified analytical goal; (iii) comprehensive, in-depth, multi-dimensional exploratory insights. Among publicly available medical datasets, The Cancer Genome Atlas (TCGA) provides various types of cancer and associated patient samples. Each sample includes tumor-associated images and paired pathology reports, which align well with our requirements. Therefore, we select it as our source data.

Our construction methodology combines mainly manual curation with LLM–assisted generation. We also performed a human review to ensure image quality and evidence validity. The overall pipeline is illustrated in Figure 1. We describe each step of the pipeline in detail as follows.

### 3.2.1 STEP 1: PATHOLOGICAL IMAGE PROCESSING

In the TCGA repository, each pathological whole-slide image (WSI) is stored as SVS. To convert WSI into suitable inputs for LMMs, we applied a standardized image processing pipeline. First, we instantiate a slide object and extract essential metadata such as pixel spacing and the dimensions of each pyramid level. Next, to preserve the global structure and large-scale morphological features while reducing the WSI to an acceptable size, we perform whole-slide downsampling. Given a target maximum output dimension, we compute an appropriate downsample ratio and select the optimal pyramid level. After color normalization, the images are exported as PNGs for downstream use.

To ensure that the final images are clear, complete and usable, we also utilize an automated check using an LMM combined with manual review. This step filters out images that are unreadable or corrupted and yields a curated set of pathological images suitable for data analysis.

### 3.2.2 STEP 2: PATHOLOGICAL REPORT PROCESSING AND INSIGHT EXTRACTION

We first retrieve the corresponding pathology reports (PDF) from the TCGA repository based on the case name of each sample. Next, we convert the reports to plain text through OCR and then inspect and correct them through LLM assistance and human verification. We inspect each report based on multiple quality criteria, including the absence of invalid characters or corrupted content, coherence of diagnostic statements, and alignment between textual descriptions and expected clinical content. Therefore, insight generation from plain-text reports is carried out in four stages as follows:

1. **Report Decomposition**: We apply an LLM to extract the key items of the report, represented as a sequence of evidence snippets that form a progressive, interrelated chain of findings.
2. **Insight Generation**: Guided by six insight types (details in Appendix A.2), we analyze each evidence and employ an LLM to generate insights. Moreover, we compute a confidence score for each insight to indicate quality, and those insights with low confidence are manually filtered.
3. **Analytical Questions Generation**: To enhance analytical depth and hierarchy, we pose goal-directed questions for each insight, ensuring a logical progression that enables incremental discovery of deeper and meaningful findings.
4. **Human Verification**: We reviewed the questions, insights, and their corresponding evidence excerpt to confirm logical consistency, factual accuracy, and rationality.

### 3.2.3 STEP 3: GOAL GENERATION

The analytical goal has two key properties: (i) it must be clearly and unambiguously stated. (ii) effectively guide both the generation of analysis questions and the overall analytical strategy. We analyze the logical relationships and dependencies among these generated questions and synthesize a concise, overarching analysis goal. To avoid hallucinations or misinterpretations, each generated goal is subject to human verification. We retained goals that are precise, coherent with the underlying questions, and appropriate to guide downstream analyses.

| (a) Quality Assessment | | | (b) Redundancy Assessment | | |
|---|---|---|---|---|---|
| Dimension | LLM Eval | Human Annotation | Metric | Questions | Insights |
| Correctness | 0.906 | 0.919 | TC Similarity | 0.0555 | 0.0307 |
| Rationality | 0.876 | 0.891 | Self-BLEU | 0.2285 | 0.0698 |
| Coherence | 0.910 | 0.930 | Distinct-2 | 0.7748 | 0.9355 |

Table 1: Data quality and redundancy analysis of MedInsightBench.

| Dataset | Input | Output | Topic & Area | Data Size | Construction Method |
|---|---|---|---|---|---|
| Spider 2.0 Lei et al. (2024) | Question | SQL Query | Enterprise-level | 632 | Machine & Human-Labeled |
| MatPlotBench Yang et al. (2024) | Question+Table | Vis Image | Data Visualization | 100 | Machine & Human-Labeled |
| InfiAgent-DABench Hu et al. (2024) | Question+Table | Answer | Data Analysis | 603 | Machine-Labeled |
| MedAgentsBench Tang et al. (2025) | Question | Answer | Clinical Analysis | 862 | Existed Dataset Combined |
| InsightBench Sahu et al. (2025) | Goal+Table | Insights | Business Analysis | 100 | Human-Labeled |
| MedInsightBench | Goal+**Image** | Insights | Medical Analysis | 332 | Machine-Labeled & Human-Verified |

Table 2: Comparison of MedInsightBench with other existing benchmarks.

## 3.3 EVALUATION FRAMEWORK

Current insights discovery evaluations are predominantly based on automated text matching metrics and G-Eval scoring, with InsightBench (Sahu et al., 2025) further narrowing the assessment to a single LLM evaluator, introducing the risk of amplifying inherent biases. Furthermore, existing protocols only compare predicted outputs with annotated ground-truth, disregarding hallucinated or incorrect predictions, while failing to identify the novel and unannotated insights. To address these limitations, we propose a refined automated evaluation framework that more accurately reflects analytical capability through four complementary metrics: **Insight Recall**, **Insight Precision**, **Insight F1-score**, and **Insight Novelty**. This approach enables explicit assessment of correct retrieval, error rates, overall balance, and discovery of previously unrecognized insights. The details of each evaluation metric are described in Appendix B.

## 3.4 DATA QUALITY ANALYSIS

To verify the quality of the dataset, we conducted an in-depth annotation across three dimensions:

- **Correctness**: whether each set of questions strictly corresponds to the stated goal and the pathological images without factual errors.
- **Rationality**: whether each insight satisfies the goal's requirements and is logically sound.
- **Coherence**: whether insights in each case are internally consistent and mutually compatible.

We randomly sampled 100 instances and annotated them by both LMM (i.e., OpenAI o3) and human experts, computing the accuracy rate for each dimension. The results are reported in Table 1a.

In addition, we evaluated redundancy for each question and insight using three metrics. First, we compute cosine similarity based on TF-IDF vector representations and average the resulting scores. Second, we computed Self-BLEU (Zhu et al., 2018) for each sentence to assess n-gram repetitiveness. Third, we measured Distinct-2, defined as the ratio of unique bigrams to total bigrams across all sentences. Generally, a higher TF-IDF cosine similarity and Self-BLEU indicate greater redundancy, while a Distinct-2 value closer to 1 reflects greater lexical diversity and lower redundancy. Table 1b reports these redundancy statistics. Through this rigorous quality assurance process, our dataset meets a high standard of reliability and scholarly validity.

## 3.5 BENCHMARK STATISTIC

The MedInsightBench dataset comprises 332 samples, each of which contains a single cancer pathology image, a specific goal, and several medical insights, yielding a total of 3,933 insights across the dataset. Each sample is annotated with one of the four difficulty levels. Furthermore, each insight is labeled with an insight category, an associated question, and an excerpt of evidence drawn from the original report. In addition, compared to other well-regarded datasets, MedInsightBench stands out for its large-scale image-modal insights, which are displayed in Table 2.

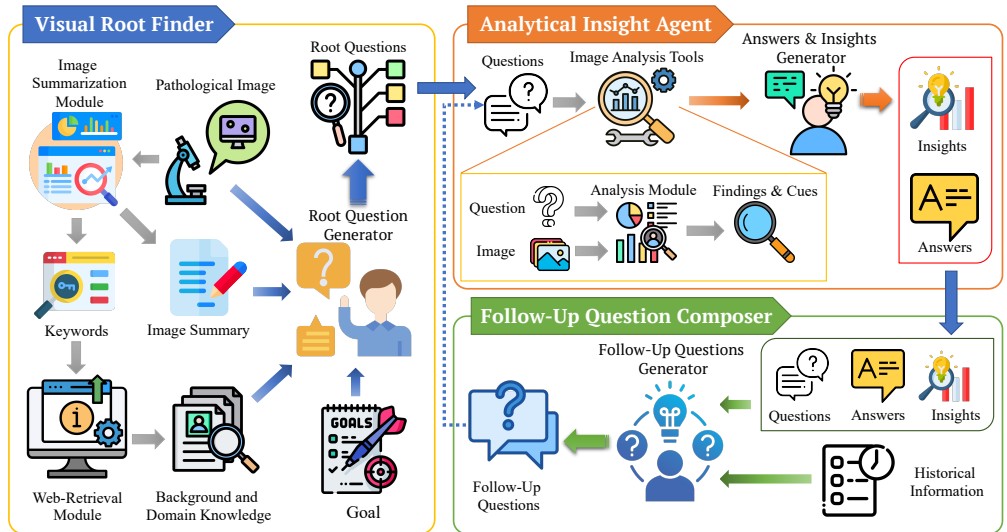

Figure 2: The overall workflow of MedInsightAgent. The framework consists of three main components: Visual Root Finder, Analytical Insight Agent, and Follow-Up Question Composer.

# 4 MEDINSIGHTAGENT: A MULTI-AGENT FRAMEWORK FOR MEDICAL INSIGHT DISCOVERY

Due to the suboptimal performance and inherent limitations of LMM in medical insight discovery tasks, we design a multi-agent framework named MedInsightAgent. The framework decomposes the insight discovery process into three specialized agents: 1) **Visual Root Finder**: Given the analytical goal, analyze the image, summarize, identify salient visual features, and generate an initial set of root questions. 2) **Analytical Insight Agent**: Answer each question using the image and associated evidence, and finally generate medical insights. 3) **Follow-up Question Composer**: Generate follow-up questions that probe deeper or explore complementary perspectives to refine and extend the discovered insights. The overall architecture is illustrated in Figure 2. We describe the processing flow and implementation details of each agent as follows.

## 4.1 VISUAL ROOT FINDER

The Visual Root Finder (VRF) takes a medical image $I$ and an analytical goal $G$ as input and generates an initial set of root questions $Q = \{q_i\}_{i=1}^m$, where $m$ is the number of questions. These root questions define the primary directions for exploration and guide subsequent insight generation.

To improve the quality of root questions, Visual Root Finder first gathers supplementary information. It incorporates two information-acquisition modules: (1) **Image-Summarization Module** $\mathcal{ISM}_{\text{img}}$. To broadly explore the medical image, the module performs an initial interpretation, extracting prominent visual features and observations $F$. Then it generates various and representative keywords $K$, formalized as $\mathcal{ISM}_{\text{img}} : I \mapsto (F, K)$. (2) **Web-Retrieval Module** $\mathcal{WRM}$. Using keywords $K$, this module retrieves domain knowledge by querying online resources (e.g., literature, reports) and returns the top ten relevant items $D = \mathcal{WRM}(K) = \{d_1, \ldots, d_{10}\}$.

Finally, the **Root Question Generator** $\mathcal{L}$ takes the medical image, the predefined analysis goal, and the retrieved information to produce a set of high-quality, precise, and concrete root questions that form the foundation for downstream analytical agents. The general process is formalized in Eq. 1.

$$\text{VRF}(I, G) = \mathcal{L}\big(I, G, F, D\big) \Rightarrow Q \tag{1}$$

## 4.2 ANALYTICAL INSIGHT AGENT

The Analytical Insight Agent (AIA) generates answers $A = \{a_i\}_{i=1}^m$ of the root questions and derives meaningful insights $S = \{s_i\}_{i=1}^m$. Since different questions probe distinct analytical facets,

directly interpreting the pathological image often leads to hallucinations or incomplete responses. Thus, it is essential to explicitly extract image evidence relevant to questions before answering them.

For this targeted evidence extraction, we employ PathGen-LLaVA (Sun et al., 2025b), which is an LMM built on the LLaVA architecture and fine-tuned on the PathGen pathology dataset (Sun et al., 2025b), as an **Image-Analysis Tool** $\mathcal{IAT}$. For each root question $q_i$, it analyzes the image $I$ and outputs relevant pathological findings and visual cues $E_i$, where $E_i = \mathcal{IAT}(I, q_i)$. These structured and question-specific image features serve as grounded evidence for subsequent reasoning.

Finally, the **Answers and Insights Generator** $\mathcal{G}$ takes the question, the pathological image, and the extracted findings to produce a rational answer and a concise, clinically meaningful insight. The overall formula is shown in Eq. 2.

$$\text{AIA}(I, Q) = \left\{ \mathcal{G}\big(q_i, I, E_i\big) \right\}_{i=1}^{m} \Rightarrow A, S \tag{2}$$

### 4.3 FOLLOW-UP QUESTION COMPOSER

The initial set of root question often suffers from coverage limitations and stochastic variability. To address this, we introduce the Follow-up Question Composer (FQC), which generates deeper and more penetrating questions for each root question. The follow-up questions $T = \{t_i\}_{i=1}^{m}$ must satisfy two criteria: (i) They must be relevant to the image $I$ and aligned with the analytical goal $G$. (ii) They must be distinct yet logically derived from the original root question, extending the inquiry to explore additional facets of the pathological image.

The **Follow-Up Question Generator** $\mathcal{F}$ first generates $n$ candidate follow-up questions $C = \{c_i\}_{i=1}^{n}$. Then the **Question Selector** $\mathcal{S}$ scores each candidate and selects the highest-scoring one $c_{best}$. The process is formalized in Eq. 3.

$$\text{FQC}(I, G, Q, A, F, D) = \mathcal{S}\Big(\big\{\mathcal{F}(I, G, q_i, a_i, F, D)\big\}_{i=1}^{n}\Big) = \mathcal{S}(C) \Rightarrow c_{best} \tag{3}$$

The selected follow-up question $c_{best}$ is then passed to the Analytical Insight Agent to generate new insights. The process is controlled by an exploration depth parameter $p$ ($p \geq 0$), which specifies the number of follow-up iteration cycles. Each root question is expanded through $p$ rounds before termination, after which the system outputs all accumulated insights. In particular, if $r$ root questions are generated, the total number of final insights ($Ins$) can be computed as $Ins = r \times (q + 1)$.

## 5 EXPERIMENTS AND ANALYSIS

### 5.1 EXPERIMENTAL SETUP

**Baselines** We evaluated the following baselines on MedInsightBench:

- **Large Multi-modal Models**: We directly utilize several LMMs including GPT-4o (OpenAI, 2024), GPT-5 (OpenAI, 2025), Deepseek-VL2 (Wu et al., 2024), Qwen2.5-VL-32B-Instruct (Bai et al., 2025) and InternVL3-38B (Chen et al., 2024b) to generate insights.
- **React Framework**: We implemented a ReAct (Yao et al., 2023) structure agent and equipped it with external tools such as a computation module and a web-search interface.
- **MedInsightAgent**: Our agent system discovers high-quality insights through an iterative loop of analysis, targeted question generation, answering, insights derivation and follow-up questioning.

**Agent Implementation details** In each agent framework, we use GPT-4o and Qwen2.5-VL-32B-Instruct as the backbone LMMs for MedInsightAgent and GPT-4o as the backbone for the ReAct framework. All LMMs are configured with a temperature of 0 to ensure deterministic output. In our MedInsightAgent, we run 4 rounds of iterations, with 3 new questions generated in each round. Similarly, the ReAct agent is set to generate the same number of questions to ensure rationality.

**Metrics** For recall and precision assessment, we employ two evaluators: ROUGE-1 (Lin, 2004) and G-Eval (Liu et al., 2023). Specifically, the G-Eval score is calculated as the average of the GPT-3.5-Turbo and Gemini 2.5 Pro scores. Next, Recall and Precision are calculated using Eq. 4 and 5,

| Baselines | Insights Recall | | Insights Precision | | Insights $F_1$ | | Insights Novelty | |
|---|---|---|---|---|---|---|---|---|
| | ROUGE-1 | G-Eval | ROUGE-1 | G-Eval | ROUGE-1 | G-Eval | Original | Innovation |
| *LMM-only* | | | | | | | | |
| GPT-4o | 0.180 | 0.298 | 0.209 | 0.358 | 0.193 | 0.325 | 0.129 | 0.209 |
| GPT-5 | 0.187 | 0.305 | 0.185 | 0.365 | 0.186 | 0.332 | 0.132 | 0.213 |
| Deepseek-VL2 | 0.183 | 0.323 | **0.228** | 0.407 | **0.203** | 0.360 | 0.196 | 0.271 |
| Qwen2.5-VL-32B-Instruct | **0.192** | **0.398** | 0.214 | **0.485** | 0.202 | **0.437** | **0.349** | **0.417** |
| InternVL3-38B | 0.177 | 0.339 | 0.201 | 0.399 | 0.188 | 0.367 | 0.161 | 0.255 |
| *Agent Framework* | | | | | | | | |
| ReAct (GPT-4o) | 0.181 | 0.302 | 0.203 | 0.371 | 0.192 | 0.332 | 0.142 | 0.224 |
| MedInsightAgent (GPT-4o) | 0.189 | 0.361 | 0.197 | 0.413 | 0.193 | 0.384 | 0.180 | 0.270 |
| MedInsightAgent (Qwen2.5-VL) | **0.212** | **0.451** | **0.209** | **0.546** | **0.211** | **0.494** | **0.416** | **0.478** |

Table 3: Insight discovery performance of different LMMs and agents on MedInsightBench. Qwen2.5-VL represents Qwen2.5-VL-32B-Instruct.

| Methods | Insights Recall | Insights Precision | Insights $F_1$ | Insights Novelty |
|---|---|---|---|---|
| Direct Decoding(GPT-4o) | 0.298 | 0.358 | 0.325 | 0.209 |
| MedInsightAgent(GPT-4o) | 0.361 | 0.413 | 0.384 | 0.270 |
|   w/o Image-Summarization Module | 0.352 | 0.407 | 0.378 | 0.253 |
|   w/o Web-Retrieval Module | 0.337 | 0.389 | 0.361 | 0.239 |
|   w/o Image-Analysis Tool | 0.331 | 0.377 | 0.353 | 0.261 |
|   w/o Follow-Up Question Composer | 0.314 | 0.365 | 0.338 | 0.233 |

Table 4: Effect of each method and module within the MedInsightAgent framework. We use the G-Eval score in Insight Recall, Precision, and $F1$ metrics, and Innovation score in Insight Novelty.

with G-Eval scores normalized for direct comparison with ROUGE-1. The final insight $F1$ score is derived using Eq. 6. Moreover, we sampled 100 data points and scored them by ten human experts. To measure insight novelty, we calculate both Original and Innovation scores using Eq. 7.

## 5.2 EXPERIMENTAL RESULTS AND FINDINGS

**Model and framework performance comparison.** Table 3 summarizes the performance of various LMMs and agent frameworks in MedInsightBench. Among LMM-only baselines, Deepseek-VL2 attains the highest ROUGE-1 score for the Insight F1 metric, while Qwen2.5-VL-32B-Instruct achieves the best G-Eval performance. Consistently, MedInsightAgent built on Qwen2.5-VL-32B-Instruct delivers the strongest overall results among all evaluated agent systems.

Insight Novelty evaluation shows that Qwen2.5-VL-32B-Instruct and its MedInsightAgent achieve the highest Innovation scores in their respective baseline groups. In addition, higher Insight F1 scores generally correspond to greater novelty. Comparison of Original and Innovation scores reveals two trends: (i) All evaluated baselines improve in Innovation relative to their Original Score. (ii) Baselines with lower Original Scores tend to exhibit larger relative gains in Innovation.

**MedInsightAgent can enhance the performance of medical insight discovery.** Comparing GPT-4o with its agent-augmented counterparts in Table 3, we observe that the ReAct framework yields only marginal improvement, whereas MedInsightAgent substantially enhances the performance of the base LMM. Furthermore, stronger base LMMs such as Qwen2.5-VL-32B-Instruct achieve even greater gains when integrated into our multi-agent pipeline.

**High precision causing redundancy and limited exploratory depth.** The Insight Precision is consistently higher than Insight Recall, suggesting that both LMMs and agents prioritize producing highly precise, well-supported insights while avoiding uncertain or exploratory outputs. Although this reduces spurious assertions, it also increases redundancy, with many high-scoring insights being repetitive. Consequently, despite the strong nominal quality of the outputs, the LMMs and agents still show limited depth and comprehensiveness of exploration.

## 5.3 ABLATION STUDY OF AGENT FRAMEWORK

Our MedInsightAgent introduces several new modules and tools that significantly enhance medical insight discovery. To assess the contribution of each component, we conducted an ablation study in Table 4. Removing any single component led to a measurable degradation in performance, under-

| Ground-Truth Insight | Output of GPT-4o | Output of MedInsightAgent(GPT-4o) |
|---|---|---|
| *Case 139*: TCGA-EJ-A7NM with Difficulty **Level 2** | | |
| Lymphovascular invasion and extensive perineural invasion suggest increased metastatic potential; consider systemic therapy evaluation. | Absence of perineural invasion in the visible sections may impact staging. | **Perineural invasion suggests a more aggressive tumor**, which might **increase the likelihood of cancer recurrence** and affect treatment decisions. |
| Multifocal extracapsular extension and seminal vesicle invasion indicate locally advanced disease with higher recurrence risk; discuss adjuvant therapy options. | Stromal desmoplasia is present, indicating possible tumor-host interaction and invasive behavior. | **A strong stromal reaction, known as desmoplasia, often indicates a more aggressive tumor** behavior and higher grade in prostate cancer, potentially impacting treatment strategies and prognosis. |
| *Case 122*: TCGA-IN-A7NT with Difficulty **Level 3** | | |
| Angiolymphatic invasion is present. | No definitive lymphovascular invasion observed, although detailed review and further sections may be warranted. | **The presence of lymphovascular invasion** can suggest a higher risk of metastasis, informing treatment decisions and prognosis. |
| Poorly differentiated (G3) adenocarcinoma suggests more aggressive tumor biology; consider comprehensive treatment approach. | There are poorly formed glands and irregular cell clustering, which may indicate a higher histologic grade. | The presence of **pronounced cellular atypia and high mitotic rate suggests a more aggressive tumor behavior**, potentially correlating with a higher risk of nodal metastasis and impacting therapeutic decisions. |
| *Case 13*: TCGA-05-4250 with Difficulty **Level 4** | | |
| Poorly differentiated grade 3 tumor with lymph node metastases, vascular invasion, and R2 resection status indicates high risk of recurrence and poor prognosis; recommend multidisciplinary discussion for adjuvant therapy consideration. | The presence of irregular nests may indicate an aggressive phenotype. | These pathological features indicate **aggressive tumor behavior**, which is crucial for **determining prognosis and guiding effective treatment strategies**. |
| Multifocal invasion of blood vessels is identified. | No clear lymphovascular invasion is observed in the current section. | **This pattern of tissue invasion suggests a higher risk of cancer spreading beyond its origin**, which could impact treatment strategies. |

Table 5: Case study of Insight Discovery. We selected three cases across different difficulty levels, with the bolded statements highlighting instances where MedInsightAgent demonstrated superior performance and the underlined parts show the defects in the output of GPT-4o.

scoring its importance. Specifically, the Image-Analysis Tool has the greatest impact on intrinsic quality metrics (Insight Recall, Precision, and F1). It provides targeted, goal-directed analysis of each slide, yielding the most relevant evidence for accurate responses. In contrast, omitting the Web-Retrieval Module results in a sharp decline in insight-novelty scores, highlighting the role of external domain knowledge and literature in fostering innovative discoveries.

Further ablations on the Follow-up Question Composer demonstrate that multi-round iteration questioning is crucial for deeper exploration and more novel insights. In general, these results confirm that the coordinated integration of image analysis, external knowledge retrieval, and iterative questioning is essential for comprehensive and innovative medical insight discovery.

### 5.4 CASE STUDY

Table 5 presents case studies of varying difficulty, comparing the ground-truth insights with outputs from GPT-4o and our MedInsightAgent (GPT-4o). The GPT-4o output often exhibits internal contradictions, incorrect judgments, and omissions of key information. In contrast, MedInsightAgent (GPT-4o) typically produces more accurate and well-grounded insights, although some outputs remain overly conceptual. These results illustrate the limitations of MedInsightBench, which demands a more domain-specific medical knowledge in the base LMMs.

### 6 CONCLUSION AND FUTURE WORK

We propose MedInsightBench, a novel benchmark for the rigorous and precise evaluation of medical-insight discovery. The benchmark supports automated evaluation and demonstrates strong concordance with human judgments. In addition, we introduce MedInsightAgent, a multi-agent framework that integrates multiple data-acquisition modules, analysis components, and external tools specifically designed for mining insights from medical images. Experimental results show that MedInsightBench exposes many key challenges in medical-insight discovery and that MedInsightAgent effectively improves the performance of several LMMs. In future work, we will further refine the multi-agent framework to improve its performance in insight discovery, thereby contributing to significant advances in medical insight research.

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

# A DATASET ANALYSIS

## A.1 DETAILED STATISTIC OF MEDINSIGHTBENCH

Figure 3 and Figure 4 present detailed statistical information on MedInsightBench, including the distribution of different Insight categories, the average number of tokens in Questions and Insights per category, and the distribution across difficulty levels.

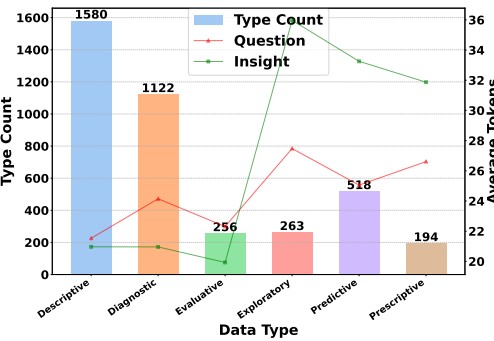
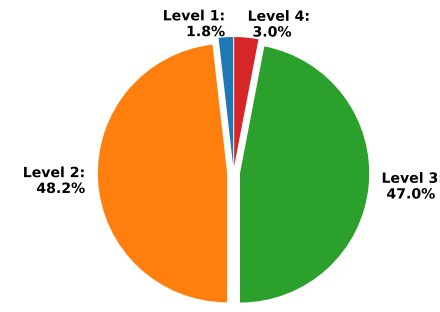

Figure 3: The distribution of different Insight Types in MedInsightBench and the average token count of Question and Insight in each type of data.

Figure 4: The distribution of different difficulty level in MedInsightBench.

## A.2 INSIGHT TYPES

In this paper, we provide a comprehensive interpretation of data insights by six insight categories. A detailed description of each insight category is provided below:

- **Descriptive**: In the medical context, descriptive insights summarize what has already occurred by aggregating and visualizing historical clinical and operational data. For example, charts of monthly inpatient admissions by diagnosis, trends in laboratory test volumes, or distributions of medication use between departments, so clinicians and administrators can quickly understand the current and past state of patients and services.

- **Diagnostic**: Diagnostic insights explain why the observed clinical or operational patterns occurred by identifying correlations, temporal associations, and plausible causal factors, such as linking a rise in postoperative infections to a change in sterilization procedures, a particular implant type, or changes in staffing, helping teams prioritize investigations and corrective actions.

- **Predictive**: Predictive insights use historical patient records, longitudinal vitals, laboratory trajectories, imaging characteristics, and social determinants to forecast future outcomes or events, such as 30-day readmission risk, likelihood of ICU transfer, or expected lab deterioration, providing probabilities and confidence estimates to inform proactive clinical planning.

- **Prescriptive**: Prescriptive insights translate predictions and diagnostics into concrete, actionable recommendations that balance benefits, risks, and constraints, for example, suggesting personalized treatment adjustments, targeted follow-up schedules, or resource allocation strategies (e.g., bed assignment or staffing changes) designed to improve results or operational efficiency.

- **Evaluative**: Evaluative insights assess the quality, reliability, and limitations of the data and analyze themselves by auditing data completeness, bias, model calibration, and external validity, for example, reporting subgroup performance disparities of a mortality model or highlighting key missing variables that undermine the conclusions.

- **Exploratory**: Exploratory insights search for unknown or unexpected patterns without a prior hypothesis, using techniques such as clustering, anomaly detection, and dimensional-

ity reduction to uncover new patient subgroups, unusual temporal events, or latent relationships, such as discovering a previously unrecognized phenotype associated with distinct biomarker patterns that merits further clinical study.

### A.3 EXAMPLES OF MEDINSIGHTBENCH

Tables 6 and 7 present Case 4 from MedInsightBench, which includes a specific goal, image of medical cancer pathology, and a series of insights. Each insight consists of a question, an insight text, and an insight type.

| Goal | Correlate histopathologic features of the tongue carcinoma with staging parameters, margin status, nodal metastasis, and HPV status to guide prognostic assessment and treatment planning. |
| --- | --- |
| Pathological Image | 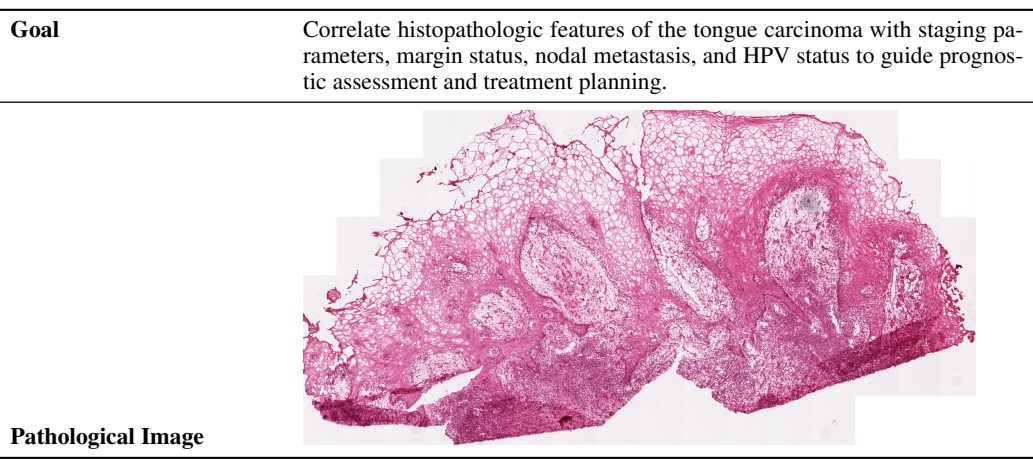 |

Table 6: Goal and pathological image of Case 4 in MedInsightBench.

## B DETAILS OF EVALUATION FRAMEWORK

At present, evaluation of insights is primarily based on automated text match metrics and G-Eval scoring. In InsightBench, it depends exclusively on LLAMA-3-Eval as the evaluator, thereby risking by model's inherent biases. Moreover, the evaluation merely measures how many ground-truth insight is matched by predicted insights while neglecting the generated erroneous insights. Lastly, it concentrates solely on discovering pre-annotated insights and does not recognize or reward the discovery of novel insights. Therefore, to address these shortcomings, we need to propose a set of revised evaluation criteria and design novel metrics accordingly.

Evaluating the medical insight discovery capabilities of the LMM and the Agent on MedInsight-Bench requires comparing the generated insights ($I$) with the annotated ground-truth insights ($GT$). To enable a more comprehensive and rigorous evaluation that accurately reflects analytical ability, we propose a novel automated evaluation framework that employs four principal measures: recall, precision, $F1$, and novelty. In the following, we detail each component of our four methodologies.

### B.1 INSIGHTS RECALL EVALUATION

To assess if ground-truth insights are discovered, we need to calculate the recall rate by adapting the iterative matching protocol. We count the scores between each ground-truth insight ($gt \in GT$) and each generated insight ($i \in I$). Then we record the highest-scoring counterpart based on each ground-truth insight ($gt$) and calculate the expectation score ($E$) as the final output. The formula for recall evaluation is shown as in equation 4, with $S$ representing the evaluator, such as ROUGE-1 or G-Eval.

$$\text{Score}_{\text{recall}}(\mathcal{S}) = E_{gt \sim \text{Unif}(GT)} \Big[ \max_{i \in I} \mathcal{S}(gt, i) \Big] \qquad (4)$$

| Insights Details |
| --- |
| Question 1: What is the diagnosis, location, size, and depth of invasion as documented in the pathology report for the excised tongue specimen?
Insight 1: Invasive keratinizing squamous cell carcinoma identified in the right lateral tongue dorsum and lateral tongue, approximately 2.0 cm in greatest dimension with muscle invasion.
Type 1: Diagnostic |
| Question 2: Does the pathology report confirm the presence of metastatic carcinoma in a lymph node from the right neck level 2, and what is the greatest dimension measurement of the identified tumor deposit?
Insight 2: Metastatic carcinoma identified in one lymph node from right neck level 2 with a tumor deposit measuring 1.9 cm.
Type 2: Diagnostic |
| Question 3: What is the status of all surgical margins regarding tumor presence in the final pathology report?
Insight 3: Final surgical margins are negative for tumor in all specimens.
Type 3: Descriptive |
| Question 4: What are the specific pathologic stage descriptors for the primary tumor, regional lymph nodes, and the number of lymph nodes examined versus involved as documented in the report?
Insight 4: Pathologic staging is pT1 pN1 with 44 lymph nodes examined and 1 involved.
Type 4: Descriptive |
| Question 5: What are the reported findings regarding histologic grade, extracapsular extension, perineural invasion, and bony/cartilage invasion?
Insight 5: Tumor is moderately-differentiated squamous cell carcinoma without extracapsular extension, perineural invasion, or bony/cartilage invasion.
Type 5: Descriptive |
| Question 6: What were the results of HPV testing for both p16 immunohistochemistry and high risk HPV in situ hybridization?
Insight 6: HPV testing performed shows p16 negative by immunohistochemistry and high risk HPV negative by in situ hybridization.
Type 6: Descriptive |
| Question 7: Does the pathology report indicate both the number of lymph nodes involved by metastatic carcinoma and the total number examined, along with the presence or absence of extracapsular extension?
Insight 7: Single lymph node metastasis (1/44) without extracapsular extension suggests intermediate recurrence risk; consider adjuvant therapy based on multidisciplinary discussion.
Type 7: Predictive |
| Question 8: Does the pathology report confirm that all intraoperative frozen section consultations for mucosal margins were benign, indicating adequate surgical clearance?
Insight 8: All intraoperative frozen section consultations for mucosal margins were benign, confirming adequate surgical clearance.
Type 8: Evaluative |
| Question 9: What are the specific benign findings and lymph node levels documented as negative for tumor in the report?
Insight 9: Additional benign findings include minor salivary gland tissue and multiple lymph node levels negative for tumor.
Type 9: Descriptive |
| Question 10: What is the HPV status of the oral cavity squamous cell carcinoma as determined by testing documented in the report?
Insight 10: HPV-negative status in an oral cavity squamous cell carcinoma suggests non-HPV driven etiology; consider additional molecular profiling for treatment guidance.
Type 10: Exploratory |

Table 7: Insight details of Case 4 in MedInsightBench.

## B.2 INSIGHTS PRECISION EVALUATION

Only focusing on the recall rate may overlook the possibility that agents generate irrelevant or unnecessary insights. To address this limitation, it is essential to further evaluate the accuracy of each generated insight to enhance the overall evaluation system. Similarly, we also enumerate the scores between the ground-truth and the generated insight. However, to calculate the precision rate, we

need to record the highest score based on each generated insight ($I$). The formula for precision evaluation is presented as in Equation 5.

$$\text{Score}_{\text{precision}}(\mathcal{S}) \; = \; E_{i \sim \text{Unif}(I)}\Big[ \max_{gt \in GT} \; \mathcal{S}(i, gt)\Big] \tag{5}$$

### B.3 Insights $F1$ Evaluation

To comprehensively assess the capability of insight discovery, we proposed a new measurement called insight $F1$ score. With the insight recall score and the insight precision score, we can calculate the insight $F1$ score through the formula in Equation 6.

$$\text{Score}_{\text{F1}}(\mathcal{S}) \; = \; \frac{2 * Score_{recall}(\mathcal{S}) * Score_{precision}(\mathcal{S})}{Score_{recall}(\mathcal{S}) + Score_{precision}(\mathcal{S})} \tag{6}$$

### B.4 Insights Novelty Evaluation

Given the limitations of merely aligning with ground-truth insights, it is essential to evaluate the capacity of discovering novel insights. We identify insights with a G-Eval score greater than 5 in the insight precision evaluation as correct, while the other insights are classified as incorrect and subjected to a secondary evaluation focused on innovation. During the evaluation, we utilize three distinct LMMs to mitigate bias. The insight can be labeled as a potential novel insight when at least two models judge it as correct. To obtain more accurate judgments, we provide LMMs with multi-modal information, including the goal, the medical image, and historical insights, and use a Chain-of-Thought (CoT) reasoning framework. The formula for novelty evaluation is expressed as in Equation 7, where $\text{LMM}_j(i) \in \{0,1\}, \quad \delta \in \{0,1\}$, $j$ is the number of LMMs, $\mathbf{1}$ means indicator function, $M$ and $N$ indicate the number of correct and incorrect insights in precision evaluation, respectively.

$$\text{Score}_{\text{novelty}} \; = \; \frac{M + \delta \sum_{i=1}^{N} \mathbf{1}\Big(\sum_{j=1}^{3} \text{LMM}_j(i) \; \geq \; 2\Big)}{N + M} \tag{7}$$

When $\delta = 1$, the formula calculates the Innovation score. For comparison, we set $\delta = 0$ to obtain the Original score during the evaluation.

## C Details of Human Annotation

Regarding annotators, our primary labelers were students with medical training. To enhance the reliability of the annotation process, we further conducted inter-annotator agreement analyses. Specifically, we collected each annotator's judgments from the dataset quality checks as well as their scoring of model outputs across all baselines, and computed several agreement metrics. The average Intraclass Correlation Coefficient (ICC) across raters was approximately 0.82, and Krippendorff's $\alpha$ was around 0.84. We additionally computed the Pearson correlation coefficient, which was about 0.76. Taken together, these results indicate that our annotators exhibit high reliability and strong consistency across individuals.

Regarding annotation details, our work includes two major components that required substantial manual annotation:

1. **Quality assessment of the benchmark dataset.** In Section 3.4, we describe in detail the metrics and procedures used for dataset quality evaluation. Each annotator reviewed the original report corresponding to each sample and performed a binary (0–1) correctness judgment on the data item or on each question/insight derived from it. The evaluation criteria included logical and medical soundness, consistency with image evidence, and alignment between the report and the ground-truth annotations.

2. **Human scoring of benchmark results**. In several experiments presented in Appendix C, we employed human experts to provide annotation-based scoring to assess the validity and reliability of our evaluation metrics. Similar to the scoring mechanism used in G-Eval, human annotators compared model- or agent-generated insights against the ground-truth insights based on semantic similarity, and rated aspects such as accuracy and plausibility on a 1–10 scale.

# D   MORE EXPERIMENT RESULTS IN DIFFERENT INSIGHT TYPE

**Insight $F1$ Score Provides a Better Reflection of Insight Capabilities.** To deepen the analysis, we also collected expert human scores and compared Insight Recall with Insight F1, which is shown in Figure 5. In particular, Insight F1 values exceed the corresponding Insight Recall scores and lie closer to human evaluations. This pattern suggests that Insight F1 is a more effective proxy for measuring medical-insight discovery capability and better reflects human judgment.

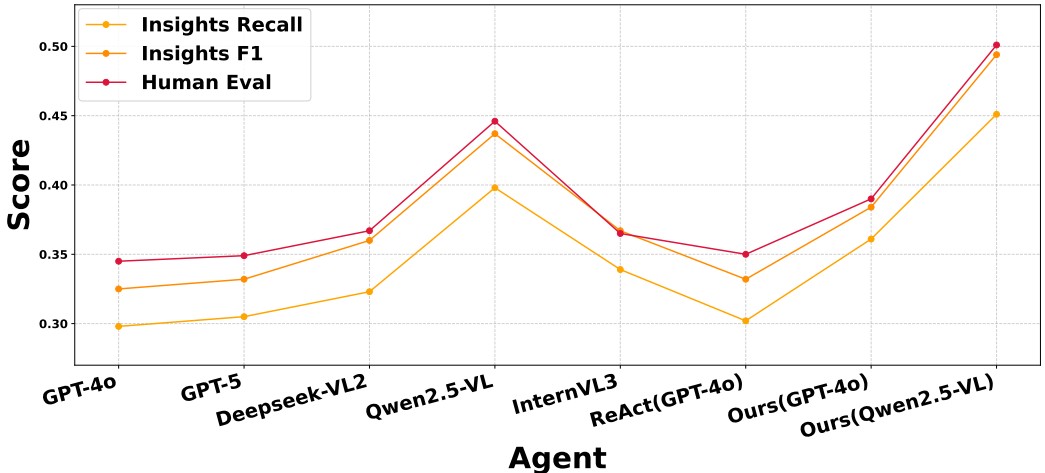

Figure 5: Comparison of G-Eval scores in Insight Recall and Insight $F1$, and Expert Scores in Human Evaluation.

**Effective scheduling and orchestration of multi-agent frameworks can yield significant performance improvements and enhanced system efficiency.** To complete the experimental comparison, we augmented the ReAct framework with the same three tools used by MedInsightAgent and evaluated its performance, which results are presented in Table 8. As shown, when provided with the same tools, ReAct indeed achieves a clear performance gain relative to the earlier setup that used only the computation module and web-search tool. However, it still falls slightly short of our proposed MedInsightAgent. This indicates that MedInsightAgent's advantage is not merely due to the use of powerful tools, but also reflects the effectiveness of the agentic orchestration mechanism we introduced.

| Baselines | Insights Recall | | Insights Precision | | Insights $F_1$ | | Insights Novelty | |
| | ROUGE-1 | G-Eval | ROUGE-1 | G-Eval | ROUGE-1 | G-Eval | Original | Innovation |
|---|---|---|---|---|---|---|---|---|
| MedInsightAgent (GPT-4o) | 0.189 | 0.361 | 0.197 | 0.413 | 0.193 | 0.384 | 0.180 | 0.270 |
| ReAct (GPT-4o) | 0.181 | 0.302 | 0.203 | 0.371 | 0.192 | 0.332 | 0.142 | 0.224 |
| with the same tools | 0.187 | 0.349 | 0.205 | 0.397 | 0.196 | 0.371 | 0.171 | 0.256 |

Table 8: Comparison of Insight discovery performance in agents with the same tools on MedInsight-Bench.

| Baselines | Insights Novelty | | |
|---|---|---|---|
| | Original | Innovation | Human Annotation |
| MedInsightAgent (GPT-4o) | 0.18 | 0.27 | 0.243 |
| MedInsightAgent (Qwen2.5-VL-32B-Instruct) | 0.416 | 0.478 | 0.451 |

Table 9: Comparison of evaluation scores in Novelty metrics between model-generated and human-assessed.

**The Novelty metric effectively reflects an agent's ability to uncover new medical insights.** To further enhance the validity and reliability of our Novelty metric, we conducted an additional round of manual inspection and sampled scoring of the model-generated insights to assess whether they contained genuine medical findings. We then compared these human-annotated scores with the automatically computed novelty scores, as shown in Table 9. We observe that the model outputs do exhibit a certain degree of misjudgment or overestimation. Nevertheless, the human-verified novelty scores still show a consistent improvement, indicating that our MedInsightAgent framework can provide discoveries that are more insightful.

# E ALGORITHM OF MEDINSIGHTAGENT

The general algorithm framework of MedInsightAgent is shown in the Algorithm 1.

---

**Algorithm 1** Overall Multi-Agent Insight Mining Framework

---

**Require:** Medical image $I$, Analysis goal $G$
**Ensure:** Final insights $S$, Answers $A$, Root questions $Q$, Follow-up questions $T$
  1: **Initialize:** $F, K, D, Q, A, S, T \leftarrow \emptyset$
      **Stage 1: Visual Root Finder (VRF)**          *(Eq. 1)*
  2: Extract initial visual summary and keywords using Image-Summarization Module:

$$(F, K) = \mathcal{IS}_M(I)$$

  3: Retrieve external domain knowledge based on keywords:

$$D = \mathcal{WR}_M(K)$$

  4: Generate initial set of root questions by combining $I$, $G$, $F$, and $D$:

$$Q = \mathcal{L}(I, G, F, D)$$

      **Stage 2: Analytical Insight Agent (AIA)**          *(Eq. 2)*
  5: **for** each root question $q_i \in Q$ **do**
  6:     Extract question-specific image evidence:

$$E_i = \mathcal{LAT}(I, q_i)$$

  7:     Generate answer and corresponding insight:

$$(a_i, s_i) = \mathcal{G}(q_i, I, E_i)$$

  8: **end for**
  9: Collect all answers and insights:

$$A = \{a_i\}_{i=1}^m, \quad S = \{s_i\}_{i=1}^m$$

      **Stage 3: Follow-up Question Composer (FQC)**          *(Eq. 3)*
10: Generate $n$ candidate follow-up questions for each root question:

$$C = \mathcal{F}(I, G, A, F, D, Q)$$

11: Select the best follow-up question $c_{\text{best}}$ using a scoring function $\mathcal{S}$:

$$c_{\text{best}} = \mathcal{S}(C)$$

12: Update follow-up question set:

$$T = T \cup \{c_{\text{best}}\}$$

      **Iteration:**
13: **while** stopping criterion not met **do**
14:     Pass $c_{\text{best}}$ back to Stage 2 (AIA) for deeper analysis
15:     Update $A$, $S$, and $Q$ with new findings
16: **end while**
17: **return** Final sets $(S, A, Q, T)$

---

# F  PROMPTS

**Prompts in Data Construction Pipeline of MedInsightBench.** Prompt 1, Prompt 2, Prompt 3 and Prompt 4 present the detailed prompts for data construction in MedInsightBench.

Prompt 1: Prompt for the Pathological Report Verification.

```
1  Given the following cancer pathology report text:
2  <report>{report_text}</report>
3
4  Instructions:
5  * Analyze the given cancer pathology report text (OCR output from a
       ↪ PDF). Perform these checks:
```

```
 6      Critical checks (must pass for the report to be usable for automated
        ↪ data-insight extraction):
 7      1. Final Diagnosis / Impression present and unambiguous (parsable
        ↪ diagnostic phrase).
 8      2. Tumor size(s) present with numeric value(s) and units (e.g., "2.3
        ↪ cm", "10 mm") or explicit statement that size not applicable.
 9      3. Tumor grade or stage info present when relevant to the specimen
        ↪ type (or explicit "not applicable").
10      4. Lymph node status present and parsable (e.g., "0/12 nodes", "3/5
        ↪ positive").
11      5. Margin status present and parsable (clear:
        ↪ positive/negative/closest margin and measurement if given).
12      6. Specimen/site and laterality clearly stated (e.g., "left lung,
        ↪ lower lobe").
13      7. Text quality/linkability: OCR not heavily garbled (no pervasive
        ↪ garbage characters), and at least one linking identifier is
        ↪ present (accession number, specimen ID, slide ID) OR the report
        ↪ contains fully parsable structured key-values that allow
        ↪ unambiguous extraction.
14
15      Additional helpful checks (not strictly required but increase
        ↪ usability):
16      8. IHC / molecular results present and include marker names with
        ↪ interpretation (e.g., "ER: positive 90%") if performed.
17      9. Clinical history/indication present (useful for context).
18      10. No internal contradictions (e.g., both "benign" and "invasive
        ↪ carcinoma" without explanation).
19      11. Negation correctly captured for critical phrases (e.g., "no
        ↪ lymphovascular invasion", "negative for malignancy").
20
21  * Decision rule:
22      - If ALL Critical checks (1 to 7) pass, output 1. Otherwise, output
        ↪ 0.
23      - NOTE: If critical checks pass but any Additional check fails,
        ↪ still output 1 but mention the missing helpful items in the reason.
24
25  * Output format requirements (strict):
26      - Your decision must be strictly enclosed in `<decision></decision>`
        ↪ tags and be either `1` or `0`.
27      - Give your reason inside `<reason></reason>`. The reason must be
        ↪ concise (max ~120 words), and must include:
28          - which critical checks passed/failed (brief labels, e.g.,
        ↪ "C1:PASS; C2:FAIL"),
29          - the top 1 to 2 specific problems found (if any), and
30          - a short recommended action (one of: "re-OCR", "manual
        ↪ pathologist review", "link accession IDs", "proceed with
        ↪ spot-checks").
31      - Your final reply must contain only these two tags and nothing else.
32
33  Refer to the example responses below.
34  Example (acceptable):
35  <decision>1</decision>
36  <reason>PASS. Critical checks: C1,C2,C3,C4,C5,C6,C7 PASS. IHC missing
        ↪ (A8). Text parsable with accession present. Recommend proceeding
        ↪ with spot-checks and include IHC curation if available.</reason>
37
38  Example (unacceptable):
39  <decision>0</decision>
40  <reason>FAIL. Critical checks failed: C2 (tumor size missing), C7 (OCR
        ↪ garbling: many non-printable chars). Recommend re-running OCR with
        ↪ an alternate engine and manual pathologist review for affected
        ↪ samples.</reason>
```

Prompt 2: Prompt for the Insights Generation.

```
1  Given the following cancer pathology report text:
2  <report>{report_text}</report>
3
4  Given the following report evidence:
5  <evidence>{evidence_text}</evidence>
6
7  Instructions:
8  * You will analyze the given cancer pathology report (OCR output that
      ↪ has passed prior usability checks) and evidence. Then extract ALL
      ↪ pathology data insights present in the report. The number of
      ↪ insights may vary by report; list every distinct insight you can
      ↪ infer from the text.
9
10 * Insight categories (choose one per insight):
11     - Descriptive: factual summaries of what the report states (e.g.,
      ↪ specimen type, tumor size, node count, IHC results).
12     - Diagnostic: statements that identify disease or etiology (e.g.,
      ↪ "invasive ductal carcinoma", "metastatic adenocarcinoma").
13     - Predictive: findings that imply future outcomes or risks (e.g.,
      ↪ "high grade and lymphovascular invasion -> increased recurrence
      ↪ risk").
14     - Prescriptive: specific, actionable recommendations based on
      ↪ findings (e.g., "recommend ER/PR testing", "suggest sentinel node
      ↪ biopsy").
15     - Evaluative: judgements about prior interventions or response
      ↪ (e.g., "treatment effect present", "no residual tumor after
      ↪ therapy").
16     - Exploratory: unexpected patterns or hypotheses worth further
      ↪ investigation (e.g., "discordant IHC vs morphology, consider
      ↪ molecular testing").
17
18 * For each insight you output, include these fields (concise,
      ↪ machine-parseable text inside the tag):
19     - Type: one of the six categories above.
20     - Insight: a concise 1 to 3 sentence paragraph that combines the
      ↪ observation (summary) and any actionable recommendation. If no
      ↪ recommendation, end with "Recommendation: none".
21     - Evidence: brief quoted text or paraphrase from the report that
      ↪ supports the insight (include enough context to locate it).
22     - Confidence: a numeric estimate from 0.0 to 1.0 reflecting how
      ↪ directly the report supports the insight.
23
24 * Output rules (strict):
25     - Each insight must be emitted as a separate
      ↪ `<insight>...</insight>` block.
26     - Inside each `insight` block, present fields in this exact order
      ↪ and simple `key: value` format (no extra markup):
27       `Type: ...; Insight: ...; Evidence: "..."; Confidence: X.X`
28     - Do NOT include any text outside the `insight` tags. Your entire
      ↪ reply must consist only of one or more `<insight>...</insight>`
      ↪ blocks and nothing else.
29     - Produce **all** insights you can extract; do not omit findings
      ↪ because they seem minor.
30
31 Refer to these examples (valid outputs):
32
33 Example - Descriptive:
34 <insight>Type: Descriptive; Insight: The specimen is a left lower
      ↪ lobectomy containing a 2.3 cm invasive adenocarcinoma;
      ↪ Recommendation: none. Evidence: "LEFT LOWER LOBECTOMY...invasive
      ↪ adenocarcinoma 2.3 cm"; Confidence: 0.95</insight>
35
36 Example - Predictive (combined):
```

```
37  <insight>Type: Predictive; Insight: High-grade morphology with
        ↪ identified lymphovascular invasion suggests increased recurrence
        ↪ risk; Recommendation: consider close surveillance and discuss
        ↪ adjuvant therapy options. Evidence: "high grade" and
        ↪ "lymphovascular invasion identified"; Confidence: 0.80</insight>
38
39  Example - Prescriptive (combined):
40  <insight>Type: Prescriptive; Insight: ER/PR status not reported while
        ↪ invasive carcinoma is present, so hormone-receptor testing is
        ↪ needed; Recommendation: order ER/PR IHC. Evidence: "ER/PR not
        ↪ reported; invasive carcinoma described"; Confidence: 0.70</insight>
```

Prompt 3: Prompt for the Questions Generation.

```
1   Given the following insight type:
2   <type>{type}</type>
3
4   Given the following insight text:
5   <insight>{insight}</insight>
6
7   Given the following Evidence in the cancer pathology report text:
8   <evidence>{evidence}</evidence>
9
10  Instructions:
11  * You will be given information on a single pathology insight (insight
        ↪ type, insight text, and corresponding Evidence excerpt from the
        ↪ cancer pathology report).
12  * Task: produce **one** clear, concise question (ending with a question
        ↪ mark) that - if answered by inspecting the original pathology
        ↪ report - would enable an analyst to derive the given insight.
13  * Constraints for the generated question:
14      - It **must end with a single question mark**.
15      - **Do not** include any verbatim text or specific phrases from the
        ↪ `Evidence` field (no quoting or restating report fragments).
16      - **Do not** perform analysis or give extraction rules inside the
        ↪ question - the question should ask *what to check* or *what
        ↪ confirmation is needed*, not how to compute it.
17      - Prefer a single sentence; be specific enough to guide an analyst
        ↪ but keep wording generic (refer to "report fields",
        ↪ "measurements", "descriptors", etc., rather than quoting report
        ↪ content).
18      - The question should be relevant to the insight's `Type`:
        ↪ Descriptive, Diagnostic, Predictive, Prescriptive, Evaluative,
        ↪ Exploratory.
19  * Output format (strict):
20      - Return exactly one `<question>...</question>` tag containing only
        ↪ the question text (nothing else).
21
22  Example output (acceptable):
23  <question>...</question>
```

Prompt 4: Prompt for the Goal Generation.

```
1   Given the following question list:
2   <question_list>{question_list}</question_list>
3
4   Instructions:
5   * You will be given a list of concise, researchable questions derived
        ↪ from pathology-report insights.
6   * Task: analyze the question list and synthesize them into a single,
        ↪ integrated Goal statement that orients image-analysis and
        ↪ downstream research efforts.
7
8   * What the Goal must do:
```

```
 9        - Capture the shared analytic direction and primary objectives
          ↪ implied by the question set (what analysts should aim to discover
          ↪ or correlate in pathology images).
10       - Be actionable at a high level (indicate the types of analyses or
          ↪ correlations to prioritize) but avoid implementation details,
          ↪ extraction rules, or step-by-step methods.
11       - Balance scope: neither overly broad nor overly detailed - enough
          ↪ to guide design of image-analysis workflows and hypothesis
          ↪ generation.
12       - Reflect clinical relevance (e.g., link morphology to
          ↪ outcome/markers, flag ambiguous cases for review) and encourage
          ↪ validation/uncertainty handling, without prescribing exact
          ↪ thresholds.
13
14 * Constraints:
15      - Produce a single paragraph, 1 to 2 sentences long (preferably 15
          ↪ to 40 words).
16      - Do NOT restate or quote the input questions; synthesize their
          ↪ themes instead.
17      - Do NOT include bullets, lists, or extra commentary.
18      - The output must be strictly enclosed in a single `<goal></goal>`
          ↪ tag and contain only that tag and the Goal text.
19
20 Example output (acceptable):
21 <goal>...</goal>
```

**Prompts in MedInsightAgent.** Prompt 5, Prompt 6, Prompt 7, Prompt 8, Prompt 9 and Prompt 10 present the detailed prompts for different parts of MedInsightAgent.

Prompt 5: Prompt for the Image Summarization Module in Visual Root Finder.

```
1 You are given a single pathology image of a cancerous tissue (H&E
    ↪ slide), and your task is to produce a concise, clinically useful
    ↪ summary describing what is seen.
2 Do not invent clinical history or definitive diagnoses beyond what the
    ↪ image supports - state uncertainty where appropriate.
3
4 Output guidance (high-level, not rigid formatting):
5 * A short summary (brief - about 1-3 sentences) describing the main
    ↪ histologic features visible at low magnification (e.g., staining,
    ↪ overall architecture, areas of increased cellularity, gland
    ↪ formation, necrosis, infiltration of surrounding tissue).
6 * A short list of 3-6 keywords highlighting the most important features.
    ↪ Each keyword must be enclosed within <keyword></keyword> tags.
7 * One brief recommendation of next steps for diagnostic confirmation
    ↪ (e.g., examine higher-power fields, perform immunohistochemistry
    ↪ panels, correlate with clinical data).
```

Prompt 6: Prompt for the Root Question Generator in Visual Root Finder.

```
1 Given the following context:
2 <context>{context}</context>
3
4 Given the following goal:
5 <goal>{goal}</goal>
6
7 Given the cancer pathology image.
8
9 Given the summary of the image:
10 <summary>{image_summary}</summary>
11
12 Given the searching results of the image summary:
13 <search_results>{search_results}</search_results>
14
```

```
15  Instructions:
16  * Write a list of questions to be solved by your cancer pathology team
        ↪ to analyze the provided cancer histopathology images and reach the
        ↪ stated goal.
17  * Focus questions on image-derived evidence (slide-level labels,
        ↪ region/patch-level features, cellular and tissue morphology, tumor
        ↪ microenvironment, staining characteristics, magnification/scale,
        ↪ annotation masks) and any linked metadata (diagnosis, clinical
        ↪ outcomes, molecular markers, patient demographics).
18  * Explore diverse aspects of the image data and metadata, and ask
        ↪ questions that are directly relevant to the goal.
19  * To better understand and analyze the cancer pathology image, you can
        ↪ refer to the given summary of the image and the search results.
20  * You must ask the right questions to surface anything interesting in
        ↪ the pathology images (morphological trends, spatial patterns, rare
        ↪ anomalies, artifacts, staining variability,
        ↪ segmentation/annotation issues, correlations with outcomes, etc.).
21  * Make sure each question can realistically be answered using the
        ↪ available data schema (image tiles/patches, labels, annotations,
        ↪ clinical/molecular metadata, quality metrics).
22  * Note that the insights your team extracts will be used to generate a
        ↪ clinical/research report.
23  * Each question should be a single-part question that requires a single
        ↪ direct answer - end the line with exactly one '?' and avoid
        ↪ compound questions.
24  * Do not number the questions.
25  * You can produce at most {max_questions} questions. Stop generating
        ↪ after that.
26  * Most importantly, each question must be enclosed within
        ↪ <question></question> tags.
27
28  Example response:
29  <question>What is the tumor status and the size of the submitted lymph
        ↪ node from the station 7 biopsy?</question>
30  <question>Does the clinical history provided in the report align with
        ↪ the pathological diagnosis regarding the specific type of
        ↪ malignancy?</question>
```

Prompt 7: Prompt for the Image Analysis Tool in Analytical Insight Agent.

```
1   ### Instruction:
2   Given the following Question:
3   <question>{question}</question>
4
5   Given: one cancer pathology image (the input image) to be inspected to
        ↪ answer the question.
6
7   Task (what to do):
8   * Analyze the image with the Question above as the analytic objective.
        ↪ Your job is to extract the **key image-derived information** that
        ↪ directly relates to answering the Question, describe the visual
        ↪ evidence, identify the image regions to inspect, note ambiguities
        ↪ or limitations, and recommend next steps (additional images,
        ↪ stains, metadata, or human review) required to confidently answer
        ↪ the Question.
9
10  Required content to produce (use these exact field names and order
        ↪ inside the output tag):
11  1. `KeyImageFindings:`  26 concise short sentences describing the
        ↪ essential visual features observed that relate to the Question
        ↪ (morphology, pattern, structures, presence/absence of features).
12  2. `RegionsOfInterest:`  brief textual description of where in the image
        ↪ the evidence appears (e.g., "upper-left field, dense invasive
        ↪ nests near adipose boundary") or integer pixel/bbox coordinates if
        ↪ available; if none, write `none`.
```

```
13  3. `Measurements:`  any quantitative values you can extract/estimate
        ↪ from the image relevant to the Question (size in mm if slide scale
        ↪ known, % area, counts); if none, write `none`.
14  4. `AmbiguitiesOrLimitations:`  concise notes on what prevents a
        ↪ definitive answer (e.g., low resolution, focal artifact, required
        ↪ IHC not visible, missing context).
15  5. `RecommendedNextSteps:`  13 short actionable recommendations to
        ↪ resolve ambiguities (e.g., request additional WSI, perform IHC for
        ↪ marker X, consult pathologist).
16  6. `Confidence:`  numeric score between 0.0 and 1.0 (one decimal place),
        ↪ estimating how confidently the image evidence supports the
        ↪ KeyImageFindings and a direct answer to the Question.
17
18  Constraints & style:
19  * Keep each field concise. Use clinical/technical wording but keep
        ↪ sentences short (one line each preferred).
20  * Assume common OCR errors and slide variability; be explicit if that
        ↪ affects interpretability.
21  * Do NOT include any narrative or extra commentary outside the required
        ↪ fields.
22
23  Output rules (strict):
24  * Your final reply must contain **only** a single
        ↪ `<findings>...</findings>` tag and nothing else.
25  * Inside the `findings` tag, present the fields in the exact order and
        ↪ format below, separated by semicolons (`;`)  no other punctuation
        ↪ structure, no newlines outside the tag:
26    `AnswerableFromImage: ...; KeyImageFindings: ...; RegionsOfInterest:
        ↪ ...; Measurements: ...; AmbiguitiesOrLimitations: ...;
        ↪ RecommendedNextSteps: ...; Confidence: X.X`
27  * All text must be replaceable by a downstream parser (avoid extra
        ↪ colons or parentheses inside field contents unless necessary).
28
29  Refer to these examples (valid outputs):
30
31  Example:
32  <findings>KeyImageFindings: Invasive glandular clusters with prominent
        ↪ nucleoli and desmoplastic stroma; RegionsOfInterest: central-right
        ↪ field near tissue edge; Measurements: largest tumor focus approx.
        ↪ 2.4 mm (estimated); AmbiguitiesOrLimitations: slide scale
        ↪ approximate, focal crush artifact; RecommendedNextSteps: confirm
        ↪ tumor size on full WSI and report scale, consider correlate with
        ↪ IHC if marker-specific question; Confidence: 0.8</findings>
```

Prompt 8: Prompt for the Answers & Insights Generator in Analytical Insight Agent.

```
1  You are trying to answer a question based on information provided. This
        ↪ is a cancer pathology image data that could potentially consist of
        ↪ interesting insights
2
3  Given the goal:
4  <goal>{goal}</goal>
5
6  Given the question:
7  <question>{question}</question>
8
9  Given the analysis (if has):
10  <analysis>{analysis}</analysis>
11
12  Given the cancer pathology image.
13
14  Instructions:
15  * Based on the analysis and other information provided above, and
        ↪ analyze the provided cancer histopathology image, please write an
        ↪ answer to the question enclosed with <question></question> tags.
```

```
16  * The answer should be a single sentence, but it should not be too
        ↪ high-level and should include the key details from the
        ↪ justification.
17  * Output must use HTML-like tags in this order: first the answer between
        ↪ <answer></answer> tags, then the justification between
        ↪ <justification></justification> tags, then the insight between
        ↪ <insight></insight> tags. Do not output any other text outside
        ↪ these tags.
18  * The justification should concisely summarize the image-derived
        ↪ evidence and any relevant linked metadata (e.g., morphology,
        ↪ cellular atypia, mitotic figures per high-power field, necrosis
        ↪ extent, spatial patterns, immunostain results, tumor fraction,
        ↪ clinical outcome) that support the answer  keep it short (13
        ↪ sentences).
19  * Use only information that can be derived from the provided
        ↪ histopathology images and linked metadata; do not invent patient
        ↪ details or data.
20  * The entire response must be factual, precise about uncertainty (if
        ↪ any), and suitable for inclusion in a clinical/research report.
21  * The insight should be a single, non-trivial, concise, and meaningful
        ↪ conclusion phrased in lay terms, grounded in the question, goal,
        ↪ and cancer histopathology image.
22  * The insight should be something interesting and grounded based on the
        ↪ question, goal, and cancer histopathology image, something that
        ↪ would be interesting.
23  * Refer to the following example response for the format of the answer,
        ↪ justification, and insight.
24
25  Example response:
26  <answer>This is a sample answer</answer>
27  <justification>This is a sample justification</justification>
28  <insight>This is a sample insight</insight>
```

Prompt 9: Prompt for the Follow-Up Question Generator in Follow-Up Question Composer.

```
1   Given the following context:
2   <context>{context}</context>
3
4   Given the following goal:
5   <goal>{goal}</goal>
6
7   Given the question and answer:
8   <question>{question}</question>
9   <answer>{answer}</answer>
10
11  Given the cancer pathology image.
12
13  Given the summary of the image:
14  <summary>{image_summary}</summary>
15
16  Given the searching results of the image summary:
17  <search_results>{search_results}</search_results>
18
19  Instructions:
20  * Produce a list of follow-up questions to explore the provided cancer
        ↪ histopathology image and reach the stated goal.
21  * Note that we have already answered the question and have the answer;
        ↪ do not include a question similar to the one above.
22  * Explore diverse aspects of the cancer histopathology image, and ask
        ↪ questions that are relevant to my goal.
23  * To better understand and analyze the cancer pathology image, you can
        ↪ refer to the given summary of the image and the search results.
24  * You must ask the right questions to surface anything interesting in
        ↪ the pathology images (morphological trends, spatial patterns, rare
```

```
         ↪ anomalies, artifacts, staining variability,
         ↪ segmentation/annotation issues, correlations with outcomes, etc.).
25 * Focus questions on image-derived evidence (slide-level labels,
         ↪ region/patch-level features, cellular and tissue morphology, tumor
         ↪ microenvironment, staining characteristics, magnification/scale,
         ↪ annotation masks) and any linked metadata (diagnosis, clinical
         ↪ outcomes, molecular markers, patient demographics).
26 * Note that the insights your team extracts will be used to generate a
         ↪ clinical/research report.
27 * Each question that you produce must be enclosed in
         ↪ <question></question> tags.
28 * Each question should be a single-part question that requires a single
         ↪ direct answer  end the line with exactly one '?' and avoid
         ↪ compound questions.
29 * Do not number the questions.
30 * You can produce at most {max_questions} questions. Stop generating
         ↪ after that.
31
32 Example response:
33 <question>What is the tumor status and the size of the submitted lymph
         ↪ node from the station 7 biopsy?</question>
34 <question>Does the clinical history provided in the report align with
         ↪ the pathological diagnosis regarding the specific type of
         ↪ malignancy?</question>
```

Prompt 10: Prompt for the Question Selector in Follow-Up Question Composer.

```
1  Given the information below:
2  <context>{context}</context>
3
4  <goal>{goal}</goal>
5
6  <prev_questions>{prev_questions_formatted}</prev_questions>
7
8  <followup_questions>{followup_questions_formatted}</followup_questions>
9
10 Instructions:
11 * Given a context and a goal, select one follow-up question from the
         ↪ above list to explore after prev_question that will help me reach
         ↪ my goal.
12 * Do not select a question similar to the previous questions above.
13 * Output only the index of the question in your response inside
         ↪ <question_id></question_id> tag.
14 * The output questions ID must be 0-indexed.
15
16 Example response:
17 <question_id>0</question_id>
```

## G    THE USE OF LARGE LANGUAGE MODELS (LLMs)

We acknowledge the use of large language models (LLMs) as auxiliary tools in the preparation of this work, primarily in the following aspects:

1. Dataset construction: During the dataset development process, we adopted an LLM-assisted approach combined with manual review. Specifically, LLMs were employed to refine and streamline the prompts used in data collection.

2. Manuscript preparation: LLMs were utilized for word choice and grammar checking, as well as for polishing the language throughout the writing of this manuscript.

3. The authors independently conceived and determined all research ideas, experimental designs, data analysis, and conclusions.

