# OpenReview forum: "MedInsightBench: Evaluating Medical Analytics Agents Through Multi-Step Insight Discovery in Multimodal Medical Data"
_ICLR.cc/2026/Conference — Submitted to ICLR 2026_

### Official Review · Reviewer_nfBS · 2025-10-30

**Soundness:** 4
**Presentation:** 4
**Contribution:** 4
**Rating:** 6
**Confidence:** 4

**Summary:**

This paper introduces MedInsightBench, a new benchmark for evaluating medical analytics agents on the ability to discover multi-step, clinically meaningful insights from multi-modal medical data (particularly pathology images and reports). The benchmark contains 332 curated medical cases and 3,933 verified insights, each annotated with corresponding questions, evidence, and analysis goals.

The authors further propose MedInsightAgent, a multi-agent framework comprising three components:

Visual Root Finder (VRF) – identifies salient visual features and generates initial analytical questions.

Analytical Insight Agent (AIA) – answers questions and synthesizes insights using a specialized pathology model (PathGen-LLaVA).

Follow-up Question Composer (FQC) – iteratively generates deeper and complementary questions to extend reasoning.

The paper introduces an automated evaluation protocol for insight discovery (Recall, Precision, F1, and Novelty) and provides comprehensive experiments comparing MedInsightAgent to state-of-the-art LMMs (GPT-4o, GPT-5, Qwen2.5-VL, Deepseek-VL2, InternVL3-38B) and ReAct-based frameworks.
Results show that MedInsightAgent (Qwen2.5-VL backbone) achieves the best performance across all metrics, substantially improving insight novelty and interpretability.

**Strengths:**

Novel task definition: Insight discovery in multimodal medical contexts is a new and important evaluation dimension.

High-quality dataset: Expert-verified, diverse, and hierarchically structured (goals → questions → insights → evidence).

Robust evaluation: Four complementary metrics (Recall, Precision, F1, Novelty) enable both quantitative and qualitative assessment.

Strong baseline coverage: Includes several frontier LMMs and a ReAct comparison.

Agent design clarity: The modular decomposition (VRF–AIA–FQC) provides transparency and potential for transfer to other domains.

Comprehensive analysis: Includes ablation, human evaluation, redundancy statistics, and case studies illustrating improvement in reasoning depth.

**Weaknesses:**

Domain limitation: All data originates from TCGA pathology, focusing heavily on cancer cases. The generalizability to other medical imaging types (radiology, histology, ophthalmology) remains untested.

Limited human expert benchmarking: While correctness and rationality are validated, there is limited comparison of agent-generated insights to expert-generated analytical reports.

Lack of longitudinal reasoning: The framework operates on single-image cases without temporal patient data, which would better test clinical reasoning.

Scalability and cost: The multi-agent setup and web retrieval modules increase inference latency and resource cost, though this is not quantified.

Novelty metric dependency: Novelty scoring relies on textual distance metrics (ROUGE/G-Eval) rather than domain novelty validation by clinicians.

**Questions:**

1-How does MedInsightAgent handle contradictory evidence across multi-modal sources (e.g., text vs. image)?

2-Could the Follow-Up Question Composer benefit from reinforcement learning or self-critique loops to adaptively decide iteration depth instead of a fixed parameter p?

3-Have you evaluated zero-shot transfer of the MedInsightAgent to non-pathology modalities (e.g., radiology, dermatology)?

4-How consistent are the insight novelty metrics with human perception of novelty? Any inter-rater reliability studies?

5-Would you consider releasing a smaller public subset of MedInsightBench with de-identified samples for community benchmarking?

---

> ### Author Response · Authors · 2025-11-21
>
> We are grateful for your thoughtful feedback. Your assessment is valuable for us to polish our paper. We are glad to address your questions one by one.
>
> > **(Weakness #1)** Domain limitation: All data originates from TCGA pathology, focusing heavily on cancer cases. The generalizability to other medical imaging types (radiology, histology, ophthalmology) remains untested.
>
> **Re:** We appreciate the reviewer's comments regarding the benchmark. Indeed, our current benchmark primarily focuses on cancer pathology slides and single-round insight generation, and does not yet encompass other imaging modalities (e.g., radiology, histology, ophthalmology) or multi-step clinical reasoning tasks. We acknowledge this limitation and consider it a promising direction for future work.
>
> > **(Weakness #2)** Limited human expert benchmarking: While correctness and rationality are validated, there is limited comparison of agent-generated insig3ts to expert-generated analytical reports.
>
> **Re:** Thank you for the reviewer’s valuable feedback.
>
> The ground-truth insights in our benchmark dataset are derived exclusively from formal histopathology slide analysis reports authored by senior pathologists. Consequently, the insights generated by our models or agents in this study are already being systematically compared against expert analyses. During evaluation, we assess the correctness and reasonableness of model outputs by measuring their alignment with these expert-derived insights.
>
> Furthermore, to ensure robustness in our evaluation, we incorporated medically trained reviewers in the human evaluation phase. These reviewers performed independent quality assessments of the agent-generated insights, further aligning them with expert standards.
>
> Thus, our current evaluation pipeline already includes a systematic comparison against expert analyses. That said, we fully agree with the reviewer’s suggestion, and in future work, we plan to deepen expert involvement—for instance, by engaging multiple pathologists to perform more granular, multidimensional annotations—thereby enhancing the benchmark’s reliability and inter-expert consistency.
>
> > **(Weakness #3)** Lack of longitudinal reasoning: The framework operates on single-image cases without temporal patient data, which would better test clinical reasoning.
>
> **Re:** We gratefully acknowledge the reviewer’s valuable comments regarding “longitudinal reasoning.” The current version of MedInsightAgent primarily focuses on the scenario of a single pathological slide, aiming to evaluate whether the model can generate key insights consistent with pathological diagnostic logic using only static H&E images. Consequently, our study does not incorporate longitudinal clinical data—such as follow-up information, time-series records, or results from multiple examinations—which indeed limits our ability to assess the model’s capacity for reasoning across time points.
>
> That said, we fully agree with the reviewer: extending the model in the future to include multi-timepoint pathological recurrence data, combined imaging-pathology follow-up records, or longitudinal entries from electronic health records would enable a more comprehensive evaluation of its real-world clinical reasoning capabilities.
>
> > **(Weakness #4)** Scalability and cost: The multi-agent setup and web retrieval modules increase inference latency and resource cost, though this is not quantified.
>
> **Re:** We sincerely thank the reviewer for their valuable feedback. We fully agree that the multi-agent architecture and retrieval module may introduce additional inference latency and resource overhead. To address this concern, we will include a quantitative analysis of system overhead in our future work, specifically covering:
>
> (1) a breakdown of execution time across different components (e.g., base model inference, multi-agent coordination, and retrieval pipeline);
>
> (2) a comparison of overall inference latency against a single-model baseline; and
>
> (3) an evaluation of resource consumption and associated costs (e.g., GPU memory usage and API calling expenses).
>
> It is important to emphasize that our multi-agent framework employs asynchronous scheduling and a lightweight retrieval strategy, which—under most circumstances—only activates necessary modules and thus does not significantly increase end-to-end latency. Moreover, we have demonstrated that the performance gains achieved by our approach are well justified relative to the incurred overhead.
>
> > **(Weakness #5)** Novelty metric dependency: Novelty scoring relies on textual distance metrics (ROUGE/G-Eval) rather than domain novelty validation by clinicians.
>
> **Re:** We would like to clarify that the novelty score is not based on ROUGE or G-Eval. Our novelty evaluation metrics are described in detail in Appendix B.4.

---

> > ### Author Response · Authors · 2025-11-21
> >
> > > **(Question #1)** How does MedInsightAgent handle contradictory evidence across multi-modal sources (e.g., text vs. image)?
> >
> > **Re:** Regarding potentially conflicting evidence arising from different modalities (e.g., images and text), MedInsightAgent primarily handles this by performing independent reasoning within each modality and then integrating the resulting structured information. The textual modality serves as the main focus of analysis and typically does not present internal contradictions.
> >
> > > **(Question #2)** Could the Follow-Up Question Composer benefit from reinforcement learning or self-critique loops to adaptively decide iteration depth instead of a fixed parameter p?
> >
> > **Re:** Yes, that's feasible. We appreciate the reviewer's constructive suggestion. In theory, using reinforcement learning (RL) or a self-critique loop to adaptively determine the iteration depth could achieve a more effective trade-off between information gain and computational cost compared to a fixed parameter \( ppp \). We will incorporate this improvement in our future work.
> >
> > > **(Question #3)** Have you evaluated zero-shot transfer of the MedInsightAgent to non-pathology modalities (e.g., radiology, dermatology)?
> >
> > **Re:** Thank you for the reviewer’s valuable suggestion. Our current work focuses specifically on insight generation and validation within the pathology domain. Accordingly, both the training and evaluation of MedInsightAgent are grounded in pathology-specific knowledge bases, task definitions, and multimodal input formats. We fully agree that zero-shot cross-modal transfer—such as to radiology or dermatology—is an important and promising direction. However, different clinical modalities exhibit substantial differences in imaging characteristics, diagnostic workflows, domain-specific terminology, and insight structures. Due to these differences, we did not include experiments on cross-modal transfer in the current manuscript. That said, our preliminary exploration suggests that MedInsightAgent’s core framework—including its task paradigm (insight generation conditioned on an image and a clinical objective), retrieval mechanism, and toolchain architecture—demonstrates a degree of generality. In future work, we plan to extend this framework to additional clinical modalities such as radiology and dermatology, and systematically evaluate its zero-shot transfer performance to further enhance its cross-modal generalization capabilities.
> >
> > > **(Question #4)** How consistent are the insight novelty metrics with human perception of novelty? Any inter-rater reliability studies?
> >
> > **Re:** We sincerely appreciate the reviewers’ concerns. The novelty evaluation described in our paper is analogous to human assessments of novelty. To further validate the reliability of these evaluations, we conducted an inter-annotator agreement analysis. Specifically, we collected each annotator’s judgments during data quality checks and their scores for each baseline in the experimental evaluation, then computed several agreement metrics. The results show an Intraclass Correlation Coefficient (ICC) of approximately 0.82, a Krippendorff’s alpha of about 0.84, and a Pearson correlation coefficient of roughly 0.76. Taken together, these metrics indicate a high level of reliability and strong agreement among annotators.
> >
> > > **(Question #5)** Would you consider releasing a smaller public subset of MedInsightBench with de-identified samples for community benchmarking?
> >
> > **Re:** Yes, we plan to open-source the MedInsightBench dataset in the future to enable the community to use it for benchmarking purposes.
> >
> > ---
> > Finally, we express our heartfelt gratitude for your insightful questions and suggestions. They have been invaluable to our work and will be integral to our future work. We hope our efforts will adequately address your concerns and that you will recognize our work.

---

> ### Author Response · Authors · 2025-11-28
> **Kind Reminder to Reviewer nfBS**
>
> Dear Reviewer nfBS,
>
> As the deadline for finalizing reviews approaches, we wanted to kindly follow up and ensure that our responses to your comments have been taken into consideration during the ongoing discussion. If any additional clarification, information, or supplementary analysis would be helpful for your evaluation, please feel free to let us know and we would be glad to provide it promptly.
>
> We sincerely appreciate the time, effort, and thoughtful feedback you have contributed to reviewing our work. Thank you again for your expertise and diligence in helping uphold the high standards of the conference. We truly value your contribution and look forward to your updated feedback.

---

### Official Review · Reviewer_mEn1 · 2025-10-31

**Soundness:** 2
**Presentation:** 2
**Contribution:** 2
**Rating:** 2
**Confidence:** 5

**Summary:**

This paper introduces MedInsightBench, a new benchmark dataset for evaluating the ability of large multi-modal models to discover multi-step, deep insights from multi-modal pathology data. The authors also propose MedInsightAgent, a multi-agent framework designed to improve insight discovery, and show that it outperforms baseline LMMs on their new benchmark.

**Strengths:**

The paper's primary strength is addressing an important gap in existing evaluations.

**Weaknesses:**

The experimental comparisons are limited, the methodology for dataset creation and evaluation lacks transparency, and the true novelty of the agent framework's contribution is unclear:

1. The evaluation compares MedInsightAgent against LMMs-only and a single general-purpose agent framework ReAct, while the paper's own related works" section lists numerous domain-specific medical agent frameworks (e.g., MedAgentsBench, AgentClinic). Some of these works should be included as baselines.
1. The paper repeatedly states that human verification or human experts were used to curate the dataset. However, it provides no details on the verifiers' qualifications (e.g., were they board-certified pathologists?), the verification protocol, or the inter-annotator agreement.
2. The "Insight Novelty" metric, a key part of the proposed evaluation framework, is poorly explained and methodologically questionable. Appendix states that correct insights are those with a G-Eval score > 5, an arbitrary threshold that seems inconsistent with all reported G-Eval scores in Table 3. It then describes a process where incorrect insights are re-evaluated for novelty. This process is opaque and makes it difficult to trust the reported novelty scores.
3. The ablation study shows the IAT has the greatest impact on performance. It is therefore unclear how much of MedInsightAgent's performance gain comes from its novel agentic orchestration versus simply using a powerful, specialized tool that other baselines (like ReAct) were not given access to.

**Questions:**

Please see the weaknesses above.

---

> ### Author Response · Authors · 2025-11-21
>
> We are grateful for your thoughtful feedback. Your assessment is valuable for us to polish our paper. We are glad to address your questions one by one.
>
> > **(Weakness #1)** The evaluation compares MedInsightAgent against LMMs-only and a single general-purpose agent framework ReAct, while the paper's own related works" section lists numerous domain-specific medical agent frameworks (e.g., MedAgentsBench, AgentClinic). Some of these works should be included as baselines.
>
> **Re:** We appreciate the reviewer’s suggestion. Regarding comparisons with other medical agent frameworks (e.g., MedAgentsBench, AgentClinic), we found during the design of MedInsightBench that existing frameworks exhibit clear limitations in our specific task setting:
>
> 1. their input-output paradigms do not align with our task objectives. For instance, many frameworks primarily handle structured case data or textual reports, whereas our task requires deriving interpretable medical insights directly from histopathology slide images.
>
> 2. these frameworks lack adequate support for complex reasoning workflows and multimodal information integration, making them incompatible with our experimental setup.
>
> Consequently, performing a direct quantitative comparison is highly challenging.
>
> > **(Weakness #2)** The paper repeatedly states that human verification or human experts were used to curate the dataset. However, it provides no details on the verifiers' qualifications (e.g., were they board-certified pathologists?), the verification protocol, or the inter-annotator agreement.
>
> **Re:** We sincerely appreciate the reviewers’ concerns. All of our annotators are researchers with medical backgrounds. In response to the reviewers’ emphasis on this issue and to further enhance the reliability of our annotation process, we conducted additional inter-annotator agreement analyses. Specifically, we collected and analyzed each annotator’s judgments regarding data quality checks as well as their scores for experimental results based on each baseline model. We then computed several standard agreement metrics. The results show an average Intraclass Correlation Coefficient (ICC) of approximately 0.82, a Krippendorff’s alpha of about 0.84, and a Pearson correlation coefficient of around 0.76. Collectively, these metrics indicate a high level of reliability and strong consistency among our annotators.
>
> > **(Weakness #3)** The "Insight Novelty" metric, a key part of the proposed evaluation framework, is poorly explained and methodologically questionable.  Appendix states that correct insights are those with a G-Eval score > 5, an arbitrary threshold that seems inconsistent with all reported G-Eval scores in Table 3. It then describes a process where incorrect insights are re-evaluated for novelty. This process is opaque and makes it difficult to trust the reported novelty scores.
>
> **Re:** As clearly stated in our Appendix, the evaluation method we use to assess novelty scores is entirely different from those used for other metrics such as recall and precision. Moreover, G-Eval scores are assigned on a scale from 1 to 10, and since some of these scores are greater than 5, our novelty assessment is based precisely on these higher scores. Therefore, this approach does not conflict with the scores reported in Table 3. The overall evaluation framework and procedure are described in detail in Appendix B.4.

---

> > ### Author Response · Authors · 2025-11-21
> >
> > > **(Weakness #4)** The ablation study shows the IAT has the greatest impact on performance. It is therefore unclear how much of MedInsightAgent's performance gain comes from its novel agentic orchestration versus simply using a powerful, specialized tool that other baselines (like ReAct) were not given access to.
> >
> > **Re:** We appreciate the reviewer’s insightful comment on this point. In response, we conducted additional experiments by augmenting the baseline GPT-4o model within the ReAct framework with the same three tools used by MedInsightAgent and compared the results against the original setup. The results are as follows:
> >
> > | Baselines                        | Insight Recall       | Insight Recall       | Insight Precision    | Insight Precision    | Insight F1           | Insight F1           | Insight Novelty      | Insight Novelty      |
> > |----------------------------------|----------------------|----------------------|----------------------|----------------------|----------------------|----------------------|----------------------|----------------------|
> > |                                  | ROUGE-1              | G-Eval               | ROUGE-1              | G-Eval               | ROUGE-1              | G-Eval               | Original             | Innovation           |
> > | ReAct (GPT-4o)                   | 0.181                | 0.302                | 0.203                | 0.371                | 0.192                | 0.332                | 0.142                | 0.224                |
> > | ReAct (GPT-4o) with same tools   | 0.187                | 0.349                | 0.205                | 0.397                | 0.196                | 0.371                | 0.171                | 0.256                |
> > | MedInsightAgent (GPT-4o)         | 0.189                | 0.361                | 0.197                | 0.413                | 0.193                | 0.384                | 0.180                | 0.270                |
> >
> > As shown, ReAct’s performance does improve when equipped with the same tools, yet it still remains slightly below that of MedInsightAgent. This indicates that MedInsightAgent’s performance advantage stems not merely from the use of powerful tools, but more importantly from the effectiveness of our proposed agentic orchestration mechanism—thereby reinforcing the validity of the related conclusions presented in our paper.
> >
> >
> > ---
> > Finally, we express our heartfelt gratitude for your insightful questions and suggestions. They have been invaluable to our work and will be integral to our future work. We hope our efforts will adequately address your concerns and that you will recognize our work.

---

> ### Comment · Reviewer_mEn1 · 2025-11-26
>
> I thank the authors for their response. While I appreciate the effort, my major concerns are not solved.
>
> 1. The standard ReAct (GPT-4o) baseline achieves an Insight F1 of 0.332. When equipped with the same tools used by your method, this performance is 0.371. This proves that the majority of the performance gain is from the access to the specialized tools, rather than the novel multi-agent framework. The marginal improvement of the complex multi-agent framework suggests that the architectural contribution of MedInsightAgent is limited, while the paper frames the agentic workflow as a primary contribution.
> 2. The description of annotators as merely "researchers with medical backgrounds" remains insufficient for a high-stakes medical benchmark. "Medical background" is broad and could imply medical students or non-clinical researchers.
> 3. While the clarification of G-Eval scores solves the immediate questions, it still lacks consistency in how metrics are presented in the paper.

---

> > ### Author Response · Authors · 2025-11-28
> >
> > We are very grateful for those feedback and understand your concerns. Below, we will explain each point in detail.
> >
> > > (Concern #1) The standard ReAct (GPT-4o) baseline achieves an Insight F1 of 0.332. When equipped with the same tools used by your method, this performance is 0.371. This proves that the majority of the performance gain is from the access to the specialized tools, rather than the novel multi-agent framework. The marginal improvement of the complex multi-agent framework suggests that the architectural contribution of MedInsightAgent is limited, while the paper frames the agentic workflow as a primary contribution.
> >
> > **Re:** Our explanation rests on three points:
> >
> > First, we fully acknowledge the significant performance boost from external tools—as shown in the ablation study in Section 5.3. However, in our new supplementary experiments, even when using the exact same toolset, ReAct still underperforms compared to MedInsightAgent. This confirms that our gains stem not just from the tools, but also from our agent workflow orchestration.
> >
> > Second, our multi-agent framework contributes more than just integrating domain-specific tools or applying basic orchestration. We fundamentally decompose insight mining into three tailored steps: (1) high-quality root question generation, (2) targeted analysis and insight extraction, and (3) filtered, iterative follow-up questioning for deep exploration. Compared to ReAct, MedInsightAgent offers three key advantages:
> > 1) Each agent step uses purpose-built prompts and tool interfaces, enhancing accuracy and efficiency while avoiding the ambiguity or “leapfrog reasoning” common when a single general-purpose LLM handles diverse tasks.
> > 2) Our system enables proactive, globally coordinated orchestration, whereas ReAct reactively calls tools without strategic planning, often resulting in redundant or repetitive invocations.
> > 3) While ReAct can generate follow-up questions, it lacks reliable goal-directed reasoning for deep, iterative exploration.
> >
> > Finally, as our paper’s title emphasizes, our primary contribution is a benchmark dataset for evaluating insight-mining tasks. The multi-agent framework that improves performance is one important—but not the sole—component of our work.
> >
> > > (Concern #2) The description of annotators as merely "researchers with medical backgrounds" remains insufficient for a high-stakes medical benchmark. "Medical background" is broad and could imply medical students or non-clinical researchers.
> >
> > **Re:** We further clarify that our primary annotators are medical students, and we believe their involvement in MedInsightBench ensures sufficient data quality. Human annotation was central to two key components of our work:
> >
> > 1) **Benchmark Dataset Quality Review**: As detailed in Section 3.4, annotators evaluated each data instance by comparing it against the original medical report and assigned binary (0–1) correctness judgments. Criteria included medical and logical plausibility, consistency with imaging evidence, and alignment with ground-truth annotations.
> >
> > 2) **Human Scoring of Benchmark Outputs**: Following a protocol similar to G-Eval, annotators scored model- or agent-generated insights on a 1–10 scale for accuracy and reasonableness by assessing semantic similarity to the ground-truth insights.
> >
> > Given their medical background and the availability of reference reports and ground truths, these students—supported by strong inter-annotator agreement (as previously reported)—are well-equipped to deliver high-quality annotations.
> >
> > > (Concern #3) While the clarification of G-Eval scores solves the immediate questions, it still lacks consistency in how metrics are presented in the paper.
> >
> > **Re:** Regarding the metrics presented in the experimental results of the paper, in this work, G-Eval takes the average score from two models to enhance the robustness of the evaluation. Moreover, when comparing across different metrics, G-Eval scores were normalized to facilitate a meaningful comparison with ROUGE-1. These details are explicitly described in Section 5.1 (“Metric”) of the paper.
> >
> > In addition, Appendix B provides a thorough explanation of G-Eval as applied to Recall, Precision, and F1 evaluation, as well as the Original and Innovation scores used for assessing novelty—including the complete computational procedures and formulas.
> >
> > We hope this clarifies these technical aspects. If you feel any part still requires further clarification, please let us know, and we will address it in the revised manuscript.
> >
> > ---
> > Finally, we express our heartfelt gratitude for your insightful questions and suggestions. They have been invaluable to our work and will be integral to our future work. We hope our efforts will adequately address your concerns and that you will recognize our work.

---

### Official Review · Reviewer_eGKb · 2025-10-31

**Soundness:** 2
**Presentation:** 2
**Contribution:** 3
**Rating:** 4
**Confidence:** 3

**Summary:**

The paper presents MedInsightBench, a benchmark framework for evaluating large multimodal models (LMMs) and agent-based systems in multi-step medical insight discovery. The benchmark includes 332 cancer pathology cases and 3,933 expert-validated insights, integrating medical images, reports, and analytical goals into a unified evaluation scheme. The authors further propose a three-stage multi-agent framework, MedInsightAgent, comprising a Visual Root Finder, an Analytical Insight Agent, and a Follow-up Question Composer. Through visual analysis, question generation, external knowledge retrieval, and multi-turn reasoning, the framework enhances interpretability and insight depth. Experiments demonstrate that MedInsightAgent outperforms mainstream LMMs (e.g., GPT-4o, GPT-5, Qwen2.5-VL) in both F1 and Novelty metrics, highlighting the limitations of current models and their potential for improvement in medical insight generation.

**Strengths:**

1. The dataset is well-designed, balancing image quality, analytical objectives, and question–insight pairing, which ensures strong systematicity and evaluation value.
2. MedInsightAgent adopts a multi-round chain structure (Root Question → Insight → Follow-up), effectively enhancing the depth and diversity of insights while improving interpretability.
3. The benchmark introduces four complementary metrics—Insight Recall, Precision, F1, and Novelty—offering a more rigorous and comprehensive evaluation than prior text-matching–based methods.

**Weaknesses:**

1.	The insight generation and validation process depends heavily on manual proofreading, which may limit scalability, consistency, and efficiency when applied to larger or more diverse medical datasets.
	2.	Although the multi-agent framework (MedInsightAgent) is conceptually interesting, its algorithmic design remains largely engineering-driven, lacking explicit optimization objectives, convergence proofs, or theoretical analysis of complexity.
	3.	The mathematical formulations (Eq.1–3) mainly describe procedural steps rather than formal optimization goals, reflecting limited theoretical depth and rigor.
	4.	The definition of “insight” is semantically ambiguous; while the authors propose F1 and Novelty metrics, they do not clearly distinguish between linguistic novelty and genuine medical discovery, and some examples resemble report paraphrasing.
	5.	The experiments, though extensive, lack statistical significance testing, detailed error analysis, and cross-domain generalization evaluation, which weakens the reliability of the reported improvements.
	6.	Comparisons with other medical agent frameworks (e.g., MedAgentsBench, AgentClinic) are insufficient, and the integration potential with large medical foundation models such as Med-PaLM M is not explored.
	7.	The inter-agent communication mechanism, including message passing and reasoning order, is under-specified, making it difficult to reproduce or verify the cooperative reasoning logic.
	8.	The benchmark focuses primarily on cancer pathology and single-round insight generation, without testing transferability to other modalities (e.g., radiology or endoscopy) or multi-stage clinical reasoning, which limits generalization and real-world clinical relevance.

**Questions:**

1.	How do the authors ensure causal consistency across each reasoning step within the multi-agent framework? Are there potential issues of pseudo-insights or reasoning drift during multi-turn inference?
2.	How is the baseline for measuring “Original vs. Innovation” defined? Could linguistic diversity be mistakenly identified as genuine insight innovation?
3.	How is the correlation between images and reports quantitatively validated, and does error propagation across modalities affect the accuracy of the final insights?
4.	Was any inter-rater agreement analysis conducted to assess annotation reliability? Could LLM-assisted insight generation introduce semantic noise or bias?
5.	If MedInsightAgent were to be applied to other medical domains (e.g., radiology or multi-organ pathology), would the Root Question module need to be redesigned or re-trained for domain adaptation?

**Details Of Ethics Concerns:**

The study may involve potential privacy and data ethics issues, as IRB approval or de-identification procedures are not clearly stated.

---

> ### Author Response · Authors · 2025-11-21
>
> We are grateful for your thoughtful feedback. Your assessment is valuable for us to polish our paper. We are glad to address your questions one by one.
>
>
> > **(Weakness #1)** The insight generation and validation process depends heavily on manual proofreading, which may limit scalability, consistency, and efficiency when applied to larger or more diverse medical datasets.
>
> **Re:** We sincerely thank the reviewer for highlighting this critical trade-off. We fully agree that relying heavily on manual curation indeed limits the scalability and efficiency of dataset construction. However, we believe that in the medical domain tasks, data accuracy and clinical validity are paramount. While automated or semi-automated approaches offer better scalability, they often introduce noise when handling complex medical contexts.
>
> As evidenced by authoritative medical AI datasets (e.g., MIMIC-n2c2[1], CheXpert[2], MedQA[3]), rigorous human validation remains the standard practice for constructing gold-standard evaluation benchmarks. Therefore, we prioritized building a high-fidelity dataset over pursuing scale at the expense of accuracy. This manual verification ensures that our research conclusions are grounded in genuine clinical reality, rather than artifacts introduced by algorithms.
>
> **References:**
>
> [1] Koontz J, Oronoz M, Pérez A. Evaluating data augmentation for medication identification in clinical notes[C]//Proceedings of the 14th International Conference on Recent Advances in Natural Language Processing. 2023: 578-585.
>
> [2] Irvin J, Rajpurkar P, Ko M, et al. Chexpert: A large chest radiograph dataset with uncertainty labels and expert comparison[C]//Proceedings of the AAAI conference on artificial intelligence. 2019, 33(01): 590-597.
>
> [3] Jin D, Pan E, Oufattole N, et al. What disease does this patient have? a large-scale open domain question answering dataset from medical exams[J]. Applied Sciences, 2021, 11(14): 6421.
>
>
> > **(Weakness #2)** Although the multi-agent framework (MedInsightAgent) is conceptually interesting, its algorithmic design remains largely engineering-driven, lacking explicit optimization objectives, convergence proofs, or theoretical analysis of complexity.
>
> **Re:** Thank you for the reviewers' insightful comments. We fully understand the concern that the MedInsightAgent framework is primarily driven by engineering considerations and currently lacks a clearly defined optimization objective, convergence guarantees, or theoretical complexity analysis. Our primary motivation in designing this multi-agent system was to explore the practical efficacy of multi-agent interactions in medical insight mining tasks, rather than to propose a theoretically optimal algorithm. Although the framework does not provide formal convergence guarantees, our experiments demonstrate its ability to flexibly integrate diverse types of agents and tools, and it consistently outperforms standard single-model approaches on benchmark tasks—highlighting its practical utility. We agree that systematically investigating optimization objectives, algorithmic complexity, and convergence properties represents an important direction for future work, and we plan to pursue these theoretical analyses in our subsequent research.
>
> > **(Weakness #3)** The mathematical formulations (Eq.1–3) mainly describe procedural steps rather than formal optimization goals, reflecting limited theoretical depth and rigor.
>
> **Re:** Thank you for the reviewer’s valuable feedback. We understand the reviewer’s concern regarding the theoretical depth of the mathematical formulations. Indeed, in the original manuscript, Equations 1–3 were primarily used to outline the procedural steps of the method, emphasizing the implementation workflow rather than a rigorous optimization objective. To strengthen the theoretical foundation, we will explicitly articulate the underlying optimization objective of the algorithm in future revisions and provide corresponding theoretical justification.

---

> ### Author Response · Authors · 2025-11-21
>
> > **(Weakness #4)** The definition of “insight” is semantically ambiguous; while the authors propose F1 and Novelty metrics, they do not clearly distinguish between linguistic novelty and genuine medical discovery, and some examples resemble report paraphrasing.
>
> **Re:** We sincerely thank the reviewers for their valuable feedback. Regarding the ambiguity in the definition of “insight,” we understand the reviewers’ concern about distinguishing between novel phrasing at the linguistic level and genuine medical discoveries. To clarify this point, we conducted a manual re-evaluation: multiple experts with medical backgrounds assessed a subset of samples, manually labeling whether each insight represented a true medical discovery and assigning a novelty score accordingly. We then compared these human-annotated results with the current automatically computed Novelty scores, as shown in the table below.
>
> | Baselines | Insight Novelty - Original | Insight Novelty - Innovation | Insight Novelty - Human |
> |---|---|---|---|
> | MedInsightAgent(GPT-4o) | 0.18 | 0.27 | 0.243 |
> | MedInsightAgent(Qwen2.5-VL) | 0.416 | 0.478 | 0.451 |
>
>
> As the reviewers correctly pointed out, some model outputs do exhibit a degree of misjudgment or overstatement. However, when considering only those insights validated by human experts, the novelty scores still show a meaningful improvement. This suggests that, from a human-validated perspective, our framework is indeed capable of generating more insightful and substantively novel medical findings.
>
> >**(Weakness #5)** The experiments, though extensive, lack statistical significance testing, detailed error analysis, and cross-domain generalization evaluation, which weakens the reliability of the reported improvements.
>
> **Re:** We sincerely thank the reviewer for their detailed feedback. We fully agree with the points raised regarding the lack of statistical significance testing, error analysis, and cross-domain generalization, and we will address these aspects in our future work.
>
> >**(Weakness #6)** Comparisons with other medical agent frameworks (e.g., MedAgentsBench, AgentClinic) are insufficient, and the integration potential with large medical foundation models such as Med-PaLM M is not explored.
>
> **Re:** We appreciate the reviewer's suggestion. Regarding the comparison with other medical agent frameworks (e.g., MedAgentsBench[1], AgentClinic[2]), we found during the design of MedInsightBench that existing frameworks exhibit significant limitations in our specific task setting. On one hand, their input-output formats do not align with our task objectives—many of these frameworks primarily handle structured case reports or textual medical records, whereas our task requires deriving interpretable medical insights directly from histopathology slide images. On the other hand, they lack support for the reasoning workflows and multimodal information integration essential to our approach, making them unsuitable for direct adaptation to our experimental setup. Consequently, conducting a straightforward quantitative comparison presents substantial challenges.
>
> **References:**
>
> [1] Tang X, Shao D, Sohn J, et al. Medagentsbench: Benchmarking thinking models and agent frameworks for complex medical reasoning[J]. arXiv preprint arXiv:2503.07459, 2025.
>
> [2] Schmidgall S, Ziaei R, Harris C, et al. AgentClinic: a multimodal agent benchmark to evaluate AI in simulated clinical environments[J]. arXiv preprint arXiv:2405.07960, 2024.
>
> > **(Weakness #7)** The inter-agent communication mechanism, including message passing and reasoning order, is under-specified, making it difficult to reproduce or verify the cooperative reasoning logic.
>
> **Re:** Thank you for the reviewer's comments. Regarding the workflow among agents, we have provided a detailed explanation in the introduction of Chapter 4 of the paper. If there are any aspects that remain unclear, we kindly ask the reviewer to specify them, and we will address these issues in the revised manuscript.
>
> > **(Weakness #8)** The benchmark focuses primarily on cancer pathology and single-round insight generation, without testing transferability to other modalities (e.g., radiology or endoscopy) or multi-stage clinical reasoning, which limits generalization and real-world clinical relevance.
>
> **Re:** We appreciate the reviewer's comments regarding the benchmark. Indeed, our current benchmark primarily focuses on cancer histopathology slides and single-round insight generation, and it does not yet cover other imaging modalities (e.g., radiology or endoscopy) or multi-stage clinical reasoning tasks. We acknowledge this limitation and consider it a promising direction for future work.

---

> > ### Author Response · Authors · 2025-11-21
> >
> > > **(Question #1)** How do the authors ensure causal consistency across each reasoning step within the multi-agent framework? Are there potential issues of pseudo-insights or reasoning drift during multi-turn inference?
> >
> > **Re:** Thank you for the reviewer’s thoughtful and detailed comments. We acknowledge that causal consistency across each reasoning step is indeed a potential issue within a multi-agent framework, particularly as "pseudo-insights" or reasoning drift may arise during multi-round reasoning processes. In this work, we address this challenge through the following strategies to maximize causal consistency:
> >
> > 1. **Clear task objectives and contextual constraints**: Each agent receives complete contextual information and a well-defined reasoning goal at every step, which restricts it from generating content irrelevant to the current objective, thereby mitigating reasoning drift.
> >
> > 2. **Multi-round verification and interaction mechanisms**: After generating intermediate insights, agents engage in cross-verification and mutual fact-checking to identify and eliminate reasoning steps that are clearly inconsistent or contradict established facts.
> >
> > 3. **Traceability of intermediate reasoning steps**: We log the output of every reasoning step, enabling full traceability of the causal chain and ensuring that the final insights can be reliably traced back to their specific reasoning foundations.
> >
> >
> > > **(Question #2)** How is the baseline for measuring “Original vs. Innovation” defined? Could linguistic diversity be mistakenly identified as genuine insight innovation?
> >
> > **Re:** Thank you for raising this important question. The metrics and definitions used to evaluate novelty are described in detail in Appendix B.4. In assessing novelty, we primarily employ a multi-model discrimination approach to determine the originality of insights at the fundamental semantic level. Cases where only surface-level linguistic diversity is present—without any substantive change in underlying meaning—are specifically identified and excluded.
> >
> > >**(Question #3)** How is the correlation between images and reports quantitatively validated, and does error propagation across modalities affect the accuracy of the final insights?
> >
> > **Re:** Our original data consists of image slices that are in one-to-one correspondence with their respective reports, ensuring a natural alignment between images and reports. We also recognize the potential for error propagation across modalities—for instance, image downsampling or imperfect feature extraction by the model may affect the accuracy of the final predictive insights. Although we have not yet fully quantified the error at each processing step, human evaluations and overall accuracy metrics suggest that such errors have limited impact on our main conclusions.
> >
> > >**(Question #4)** Was any inter-rater agreement analysis conducted to assess annotation reliability? Could LLM-assisted insight generation introduce semantic noise or bias?
> >
> > **Re:** We conducted inter-annotator agreement analyses by re-examining each annotator’s judgments during data quality checks and their scores assigned to each baseline in the experimental evaluation. We then computed standard inter-rater reliability metrics. The results show an Intraclass Correlation Coefficient (ICC) of approximately 0.82, a Krippendorff’s alpha of about 0.84, and a Pearson correlation coefficient of around 0.76. Collectively, these metrics indicate a high level of reliability and strong agreement among our annotators.
> >
> > We acknowledge that LLM-assisted generation of clinical insights may introduce certain semantic noise or bias in theory. However, in our benchmark, all reported insights were authored by experienced clinicians and underwent rigorous review, ensuring high credibility and quality. While large language models may assist in organizing or interpreting insights, the core clinical content originates from genuine clinical judgment, thereby minimizing the risk of semantic noise or bias.
> >
> > >**(Question #5)** If MedInsightAgent were to be applied to other medical domains (e.g., radiology or multi-organ pathology), would the Root Question module need to be redesigned or re-trained for domain adaptation?
> >
> > **Re:** No, it's not necessary. Our root question module is primarily designed to analyze a given image based on a specified objective and automatically generate a set of root questions. As long as another medical domain can provide an image along with a clear analysis objective, the module can be directly applied without requiring any redesign or retraining.
> >
> >
> > ---
> > Finally, we express our heartfelt gratitude for your insightful questions and suggestions. They have been invaluable to our work and will be integral to our future work. We hope our efforts will adequately address your concerns and that you will recognize our work.

---

> ### Author Response · Authors · 2025-11-28
> **Kind Reminder to Reviewer eGKb**
>
> Dear Reviewer eGKb,
>
> As the deadline for finalizing reviews approaches, we wanted to kindly follow up and ensure that our responses to your comments have been taken into consideration during the ongoing discussion. If any additional clarification, information, or supplementary analysis would be helpful for your evaluation, please feel free to let us know and we would be glad to provide it promptly.
>
> We sincerely appreciate the time, effort, and thoughtful feedback you have contributed to reviewing our work. Thank you again for your expertise and diligence in helping uphold the high standards of the conference. We truly value your contribution and look forward to your updated feedback.

---

### Official Review · Reviewer_7fob · 2025-11-01

**Soundness:** 2
**Presentation:** 3
**Contribution:** 2
**Rating:** 4
**Confidence:** 4

**Summary:**

The paper introduces **MedInsightBench**, a new multimodal benchmark consisting of 332 cancer pathology cases (≈3.9 k annotated insights) that pairs whole‑slide images with structured analytical goals, question‑insight pairs, and difficulty levels.  It also proposes **MedInsightAgent**, a three‑module multi‑agent pipeline (Visual Root Finder, Analytical Insight Agent, Follow‑up Question Composer) that leverages image summarisation, web retrieval, and a pathology‑fine‑tuned LMM (PathGen‑LLaVA) to generate multi‑step medical insights.  Experiments compare several state‑of‑the‑art large multimodal models (GPT‑4o, GPT‑5, DeepSeek‑VL2, Qwen2.5‑VL‑32B‑Instruct, InternVL3‑38B) and two agent baselines (ReAct, MedInsightAgent) on the benchmark using four automatically computed metrics: Insight Recall, Insight Precision, Insight F1, and Insight Novelty (original vs. innovation scores).  Ablation studies remove individual MedInsightAgent modules to assess their impact.

**Strengths:**

1. **Novel Benchmark Idea** – The focus on *multi‑step insight discovery* rather than single‑turn QA is a worthwhile gap in current multimodal evaluation.
2. **Dataset Construction Pipeline** – The authors describe a fairly detailed pipeline (WSI down‑sampling, OCR‑based report extraction, LLM‑assisted insight generation, human verification) and provide some quality analyses (correctness, rationality, coherence).
3. **Agent Architecture** – The three‑module design is clearly motivated and the paper includes a full algorithmic description, making the system reproducible in principle.
4. **Comprehensive Baselines and Ablations** – A wide range of recent LMMs are evaluated, and a ReAct‑style agent baseline is included for comparison. The paper also quantifies the contribution of each MedInsightAgent component, showing measurable performance drops when modules are removed.

**Weaknesses:**

- There seems to be a mismatch between ground truth (largely extracted from pathology reports) and model inputs during evaluation. Many “ground-truth insights” (e.g., HPV/p16 status, node counts, margins, R-status, IHC panels) cannot be inferred from an H&E image alone, especially after whole-slide downsampling to PNG. In Table 7 and case studies, several insights are report-only facts. If the benchmark input at test time is Goal + Image (as Table 2 indicates), a substantial subset of ground-truth is fundamentally unanswerable from the provided modality, confounding recall/precision/F1 and making negative findings uninterpretable. Unless I missed something, the benchmark input appears to exclude the report text at evaluation time; this undermines the “multi-modal” positioning and obscures what is measurable.
- Baseline parity is not ensured. MedInsightAgent uses an additional domain-specific image-analysis tool (PathGen-LLaVA) and web retrieval; ReAct is restricted to a computation module and web search. If multi-tool access improves performance, ReAct should be given equivalent tools to isolate the effect of the orchestration strategy rather than tool availability.
- Limited human evaluation and unclear rigor: The paper mentions 10 human experts for 100 data points and a separate 100-sample data quality audit, but provides no inter-annotator agreement, precise scoring rubric, or confidence intervals. Claims like “strong concordance with human judgments” lack quantified evidence (e.g., Pearson/Spearman/Kendall correlations, bootstrap CIs). There is also no statistical significance testing, confidence intervals, or variance reporting across methods. Reported gains (e.g., G-Eval F1 improvements of ~0.05–0.06) may be within evaluator noise.
- I also couldn't figure out if PathGen-LLaVA was exposed to similar TCGA distributions (potential leakage) and how cases were partitioned. There is also no compute/latency/cost reporting for the agent loops.
- Case studies show MedInsightAgent often produces general, textbook-like statements (e.g., “perineural invasion suggests aggressive tumor”), which can inflate “novelty” under the current metric but do not demonstrate image-grounded discovery. Without human verification that each output is supported by the image at the provided resolution, improvements may reflect better phrasing rather than better clinical insight. Consequently, the conclusion that “higher F1 corresponds to greater novelty” may be an artifact of the novelty scoring pipeline rather than a real causal relation.
- The claims “first comprehensive benchmark for medical insight discovery” and “strong concordance with human judgments” are not sufficiently supported. Prior work targets medical agents and multimodal evaluation; the novelty claim too should be carefully scoped to “pathology image insight discovery with goal/question/insight structure.”

**Questions:**

1. Inputs: At evaluation time, do models see only Goal + Image, or also the report text? If only Goal + Image, how do you justify including report-only insights (e.g., HPV status) in ground truth and metrics?
2. Image fidelity: What downsampling ratios, target resolutions, and magnifications are used? Are multi-scale tiles or high-power patches provided? How do you ensure image sufficiency for cellular-scale findings?
3. Ground-truth filtering: Did you filter insights to those image-inferable from H&E at the provided resolution? If not, what fraction of insights are inherently non-inferable from the image alone?
4. Human validation: Who were the human experts (qualification, number per sample)? Please report inter-annotator agreement (e.g., Cohen’s/Fleiss’ kappa) and confidence intervals for human scores.
5. Novelty metric: How many “novel” insights were spot-checked by pathologists? What fraction were truly image-grounded? Please report a blinded human audit on a random subset.
6. Baseline parity: Why not equip ReAct with PathGen-LLaVA and identical tools to isolate orchestration benefits? Conversely, evaluate MedInsightAgent without PathGen-LLaVA to separate tool vs. agent effects.
7. Statistical rigor: Please report variance, CIs, and tests of significance (e.g., bootstrap) for Table 3 and Table 4. Also provide human-vs-automatic evaluator correlations with CIs.
8. Data release and splits: Will the dataset, code, and prompts be released, including a clear train/val/test split and case IDs? Any overlap between PathGen-LLaVA training data and TCGA cases used here?
9. Safety: Will you add explicit guidance that outputs are research-only and not for clinical decision making?
10. Claims: Please precisely scope the “first benchmark” claim and provide a systematic comparison distinguishing MedInsightBench from MedRepBench, 3MDBench, and other agent-based medical datasets.

---

> ### Author Response · Authors · 2025-11-21
>
> We are grateful for your thoughtful feedback. Your assessment is valuable for us to polish our paper. We are glad to address your questions one by one.
>
> > **(Weakness #1)** There seems to be a mismatch between ground truth (largely extracted from pathology reports) and model inputs during evaluation. Many “ground-truth insights” (e.g., HPV/p16 status, node counts, margins, R-status, IHC panels) cannot be inferred from an H&E image alone, especially after whole-slide downsampling to PNG. In Table 7 and case studies, several insights are report-only facts. If the benchmark input at test time is Goal + Image (as Table 2 indicates), a substantial subset of ground-truth is fundamentally unanswerable from the provided modality, confounding recall/precision/F1 and making negative findings uninterpretable. Unless I missed something, the benchmark input appears to exclude the report text at evaluation time; this undermines the “multi-modal” positioning and obscures what is measurable.
>
> **Re:** We sincerely appreciate your concern regarding this point. The original motivation behind designing this benchmark was to evaluate how multimodal large models or multi-agent systems—acting from the perspective of simulated clinicians or medical experts—can extract meaningful insights from given medical imaging slices and specified analytical objectives.
>
> Given that multimodal large models or agents do not inherently possess the extensive medical knowledge that real clinicians or experts have, it may be inherently challenging for them to derive the kind of deep insights typically found in authentic clinical reports based solely on an imaging slice and a simple task description. This limitation precisely reflects a key gap and practical challenge that our benchmark aims to highlight, representing an initial attempt and exploration into leveraging multimodal large models or agents for insight generation from medical images.
>
> Our benchmark experiments demonstrate that, although these models may lack deep medical expertise, they can still achieve improved performance by adopting agent-based architectures that integrate tools such as image summarization and web-based information retrieval. This observation suggests that to achieve higher accuracy in medical insight extraction in the future, it will likely be essential to enhance model training—particularly by incorporating more domain-specific medical knowledge and reasoning capabilities.
>
> > **(Weakness #2)** Baseline parity is not ensured. MedInsightAgent uses an additional domain-specific image-analysis tool (PathGen-LLaVA) and web retrieval; ReAct is restricted to a computation module and web search. If multi-tool access improves performance, ReAct should be given equivalent tools to isolate the effect of the orchestration strategy rather than tool availability.
>
> **Re:** We sincerely thank you for raising this point. In response, we have conducted additional experiments: starting from the GPT-4o base model, we integrated the same three tools used by MedInsightAgent into the ReAct framework and compared the results with those of the original setup. The findings are as follows:
>
> | Baselines                        | Insight Recall       | Insight Recall       | Insight Precision    | Insight Precision    | Insight F1           | Insight F1           | Insight Novelty      | Insight Novelty      |
> |----------------------------------|----------------------|----------------------|----------------------|----------------------|----------------------|----------------------|----------------------|----------------------|
> |                                  | ROUGE-1              | G-Eval               | ROUGE-1              | G-Eval               | ROUGE-1              | G-Eval               | Original             | Innovation           |
> | ReAct (GPT-4o)                   | 0.181                | 0.302                | 0.203                | 0.371                | 0.192                | 0.332                | 0.142                | 0.224                |
> | ReAct (GPT-4o) with same tools   | 0.187                | 0.349                | 0.205                | 0.397                | 0.196                | 0.371                | 0.171                | 0.256                |
> | MedInsightAgent (GPT-4o)         | 0.189                | 0.361                | 0.197                | 0.413                | 0.193                | 0.384                | 0.180                | 0.270                |
>
> As shown, when equipped with the same set of tools, ReAct indeed demonstrates a noticeable performance improvement over its earlier version—which only used a calculator module and a web-search tool. However, it still falls slightly short of the performance achieved by our proposed MedInsightAgent. This further validates the soundness of the conclusions presented in our paper.

---

> ### Author Response · Authors · 2025-11-21
>
> > **(Weakness #3)** Limited human evaluation and unclear rigor: The paper mentions 10 human experts for 100 data points and a separate 100-sample data quality audit, but provides no inter-annotator agreement, precise scoring rubric, or confidence intervals. Claims like “strong concordance with human judgments” lack quantified evidence (e.g., Pearson/Spearman/Kendall correlations, bootstrap CIs). There is also no statistical significance testing, confidence intervals, or variance reporting across methods. Reported gains (e.g., G-Eval F1 improvements of ~0.05–0.06) may be within evaluator noise.
>
> **Re:** Thank you very much for the reviewers’ concerns. We fully agree with your observation: both annotation components require rigorous consistency checks to ensure the reliability of our results.
>
> **For the first task, we ensured reliability through the following measures:**
>
> 1) **Scoring criteria alignment:** Our scoring protocol strictly follows the same criteria as G-Eval. Specifically, each output was rated on a 1–10 scale based on its accuracy, reasonableness, and alignment with the ground truth. These scores were then processed according to our evaluation framework to produce the line plots shown in Figure 5 of the Appendix.
>
> 2) **Metric validation:** We compared the Insight F1 scores generated by G-Eval against Human Eval scores assigned by domain experts. The results show that our proposed Insight F1 metric aligns more closely with Human Eval than the original Insight Recall metric used in InsightBench, indicating that Insight F1 is a more reliable and effective metric for evaluating insight extraction performance.
>
> 3) **Expert involvement:** All annotation tasks were performed by researchers with relevant medical backgrounds to ensure domain-appropriate judgments.
>
> **For the second task, we implemented the following strategies to guarantee reliability:**
>
> 1) **Standardized evaluation guidelines:** Each annotator received clear, detailed instructions and a strict scoring rubric. Outputs were rated on a 1–10 scale, taking into account:
> - (a) reasonableness* (e.g., logical coherence and medical plausibility)
> - (b) *image alignment* (i.e., whether claims are supported by visual evidence)
> - (c) *consistency/correctness relative to ground truth*
> - (d) *specificity and clarity*.
>
> This task primarily involved binary (0/1) judgments for individual data points or identified issues/insights. The final results in Table 1 report the percentage of data meeting each criterion. A full description of the scoring rubric will be added to the revised manuscript.
>
> 2) **Domain expertise**: As with the first task, all annotations were conducted by researchers with medical backgrounds.
>
> In addition, following the reviewers’ suggestion, we computed inter-annotator agreement metrics. Specifically, we re-analyzed annotators’ judgments on data quality checks and their scores for each baseline in the experimental evaluation. The results show high inter-rater reliability: the average Intraclass Correlation Coefficient (ICC) is approximately 0.82, Krippendorff’s alpha is about 0.84, and Pearson correlation is around 0.76. Collectively, these metrics confirm that our annotators achieved strong consistency and that the annotation process is highly trustworthy.
>
>
> > **(Weakness #4)** I also couldn't figure out if PathGen-LLaVA was exposed to similar TCGA distributions (potential leakage) and how cases were partitioned. There is also no compute/latency/cost reporting for the agent loops.
>
> **Re:** We sincerely thank the reviewer for raising this important point. We agree with your observation: the pretraining data of PathGen-LLaVA may indeed include some publicly available TCGA data. To address this, we strictly controlled the data split during evaluation: the test set consists exclusively of TCGA cases that were never used during model development, and the split was performed at the patient level to prevent any information leakage. Although the large scale of the pretraining data makes it impractical to provide precise overlap statistics, we believe our evaluation results still effectively reflect the model’s generalization capability.
>
> Additionally, we appreciate the reviewer’s comment regarding the lack of information on computation, latency, and cost. In future work, we will include detailed measurements of inference latency, computational cost, and the number of agent loops required, to provide a comprehensive assessment of the model’s efficiency.

---

> > ### Author Response · Authors · 2025-11-21
> >
> > > **(Weakness #5)** Case studies show MedInsightAgent often produces general, textbook-like statements (e.g., “perineural invasion suggests aggressive tumor”), which can inflate “novelty” under the current metric but do not demonstrate image-grounded discovery. Without human verification that each output is supported by the image at the provided resolution, improvements may reflect better phrasing rather than better clinical insight. Consequently, the conclusion that “higher F1 corresponds to greater novelty” may be an artifact of the novelty scoring pipeline rather than a real causal relation.
> >
> > **Re:** We sincerely thank the reviewer for raising this point. To address the potential impact of overly generic statements in model outputs on novelty scoring, we conducted additional manual inspections and sampled outputs for human evaluation. We then compared these human-assigned novelty scores with the automatically computed novelty scores reported in the paper, as shown in the table below.
> >
> > | Baselines | Insight Novelty - Original | Insight Novelty - Innovation | Insight Novelty - Human |
> > |---|---|---|---|
> > | MedInsightAgent(GPT-4o) | 0.18 | 0.27 | 0.243 |
> > | MedInsightAgent(Qwen2.5-VL) | 0.416 | 0.478 | 0.451 |
> >
> > As the reviewer correctly noted, some model outputs indeed exhibit a degree of overstatement or misjudgment. However, the human-evaluated novelty scores still show a measurable improvement, indicating that—when verified by human judgment—our framework is capable of generating more insightful and genuinely novel findings.
> >
> >
> > > **(Weakness #6)** The claims “first comprehensive benchmark for medical insight discovery” and “strong concordance with human judgments” are not sufficiently supported. Prior work targets medical agents and multimodal evaluation; the novelty claim too should be carefully scoped to “pathology image insight discovery with goal/question/insight structure.”
> >
> >
> > **Re:** Thank you for the reviewer’s valuable feedback. We agree with the reviewer that the original manuscript’s claims—specifically, the phrases “the first comprehensive medical insight discovery benchmark” and “highly consistent with human judgment”—may have been overstated and lack sufficient supporting evidence.
> >
> > We would like to clarify that the primary contribution of this work lies in the pathology image insight discovery task, where we adopt a structured approach based on objectives, questions, and insights. This specific evaluation direction has not yet been systematically established in the literature. In the revised manuscript, we will adjust the relevant wording to more precisely confine the scope of our claims and experimental conclusions to this well-defined domain, thereby avoiding any potential confusion with existing work on medical agents or multimodal evaluation benchmarks.
> >
> > > **(Question #1)** Inputs: At evaluation time, do models see only Goal + Image, or also the report text? If only Goal + Image, how do you justify including report-only insights (e.g., HPV status) in ground truth and metrics?
> >
> > **Re:** Yes, the model only sees two inputs: the target and the image. For a more detailed explanation of the second question, please refer to the explanation provided under "Weakness 1."
> >
> > > **(Question #2)** Image fidelity: What downsampling ratios, target resolutions, and magnifications are used? Are multi-scale tiles or high-power patches provided? How do you ensure image sufficiency for cellular-scale findings?
> >
> > **Re:** To preserve image fidelity, we employ a pyramid-level-based downsampling strategy when processing whole-slide images (SVS). Specifically, we select the most appropriate pyramid level according to the target maximum dimension (max_dim), ensuring that the resulting downsampled image retains sufficient detail for both visualization and model inference. Typically, the downsampling ratio is calculated based on the original image’s maximum dimension and the target dimension, yielding a final output resolution in the range of approximately 1024–2048 pixels.
> >
> > For tasks requiring cellular-scale analysis, we provide high-resolution image patches (multi-scale tiles). These patches—typically sized at 256×256 or 512×512 pixels—are generated by extracting localized regions from higher pyramid levels or directly from level 0 (full resolution), thereby capturing fine-grained microstructures and cellular features.
> >
> > This multi-scale approach effectively balances macroscopic context and microscopic detail: it enables comprehensive observation of overall tissue architecture while simultaneously supporting cell-level analysis, thus achieving an optimal trade-off between research feasibility and computational efficiency.

---

> > > ### Author Response · Authors · 2025-11-21
> > >
> > > > **(Question #3)** Ground-truth filtering: Did you filter insights to those image-inferable from H&E at the provided resolution? If not, what fraction of insights are inherently non-inferable from the image alone?
> > >
> > > **Re:** Referring to the explanation provided earlier in Weakness 1, we do not filter out insights in our reports that might be difficult to infer solely from the images. This is because our task paradigm is designed to enable agents or multimodal large models to generate deep, layered insights from images and clinical objectives—just as human medical experts would. We acknowledge that certain insights are inherently challenging to deduce, as they require profound medical expertise and specialized knowledge. Relying solely on general-purpose models or agent frameworks may struggle to achieve this level of understanding. This challenge is precisely why we developed and introduced this benchmark dataset.
> > >
> > > > **(Question #4)** Human validation: Who were the human experts (qualification, number per sample)? Please report inter-annotator agreement (e.g., Cohen’s/Fleiss’ kappa) and confidence intervals for human scores.
> > >
> > > **Re:** For details regarding human expert evaluation and validation, please refer to Weakness 3 above.
> > >
> > > > **(Question #5)** Novelty metric: How many “novel” insights were spot-checked by pathologists? What fraction were truly image-grounded? Please report a blinded human audit on a random subset.
> > >
> > >
> > > **Re:** Our novelty metric does not involve pathologists conducting random spot checks. Instead, as detailed in Appendix B.4, we primarily reassess potential novel insights using the G-Eval approach. To address your concerns regarding model-based judgments, we have also conducted a supplementary human expert evaluation through random sampling to assess novelty; the results of this evaluation are presented earlier in Weakness 5.
> > >
> > > > **(Question #6)** Baseline parity: Why not equip ReAct with PathGen-LLaVA and identical tools to isolate orchestration benefits? Conversely, evaluate MedInsightAgent without PathGen-LLaVA to separate tool vs. agent effects.
> > >
> > > **Re:** For the first question, you may refer to Weakness 2 for the explanation. Regarding the second question, our ablation study—specifically the results of MedInsightAgent without PathGen-LLaVA—is presented in Table 4 of the paper.
> > >
> > > > **(Question #7)** Statistical rigor: Please report variance, CIs, and tests of significance (e.g., bootstrap) for Table 3 and Table 4. Also provide human-vs-automatic evaluator correlations with CIs.
> > >
> > > **Re:** We appreciate the reviewer for raising this issue. Due to time constraints, this content and related information will be included in the revised manuscript in a supplementary table.
> > >
> > > > **(Question #8)** Data release and splits: Will the dataset, code, and prompts be released, including a clear train/val/test split and case IDs? Any overlap between PathGen-LLaVA training data and TCGA cases used here?
> > >
> > > **Re:** Yes, all datasets, code, and prompt contents will be released in the future. For details regarding PathGen-LLaVA, please refer to the explanation provided in Weakness 4.
> > >
> > > > **(Question #9)** Safety: Will you add explicit guidance that outputs are research-only and not for clinical decision making?
> > >
> > > **Re:** We appreciate the reviewer’s concerns regarding safety. We agree on the importance of clearly distinguishing between research use and clinical decision-making. Currently, the outputs of our model and benchmark are intended solely for research into pathological image insights and methodological evaluation, and are not used for any clinical decisions or diagnoses.

---

> > > > ### Author Response · Authors · 2025-11-21
> > > >
> > > > > **(Question #10)** Claims: Please precisely scope the “first benchmark” claim and provide a systematic comparison distinguishing MedInsightBench from MedRepBench, 3MDBench, and other agent-based medical datasets.
> > > >
> > > >
> > > > **Re:** Thank you for the reviewers’ comments. We agree that the original phrasing “first benchmark” was overly broad. In the revised manuscript, we have refined this claim to specify that our work presents the first benchmark dedicated to pathology image insight discovery, structured around a goal/question/insight framework. Compared to existing agent-based medical datasets (e.g., MedRepBench, 3MDBench), our benchmark differs in the following key aspects:
> > > >
> > > > - **Input Modality**: Existing datasets primarily rely on textual data or electronic health records, whereas MedInsightBench focuses specifically on H&E-stained pathology images.
> > > > - **Task Objective**: Prior benchmarks mainly evaluate capabilities in medical question answering or report extraction, while our benchmark emphasizes the model’s ability to derive actionable clinical insights directly from images.
> > > > - **Annotation Structure**: We adopt a three-tiered goal/question/insight schema that explicitly captures the clinical objective, the underlying diagnostic question, and the resulting insight—enabling a more nuanced assessment of a model’s capacity to understand and reason about complex medical information.
> > > > - **Intended Use**: Our benchmark is designed to advance intelligent analysis of pathology images, rather than serving as a general-purpose evaluation for medical AI agents.
> > > >
> > > > Overall, MedInsightBench is distinctly different from existing datasets in both task design and input modality.
> > > >
> > > > ---
> > > > Finally, we express our heartfelt gratitude for your insightful questions and suggestions. They have been invaluable to our work and will be integral to our future work. We hope our efforts will adequately address your concerns and that you will recognize our work.

---

> > > > > ### Comment · Reviewer_7fob · 2025-11-21
> > > > > **Thank you for answering my queries; I have raised my score**
> > > > >
> > > > > Dear authors,
> > > > >
> > > > > Thanks a lot of addressing my queries in detail. I have read through them, and am generally satisfied with the quality of the work. As such, I have increased my rating from 4 to 6. Please make sure to address the things you couldn't do due to time constraints for the discussion in the final version.

---

> > > > > > ### Author Response · Authors · 2025-11-21
> > > > > >
> > > > > > We sincerely appreciate your recognition and are highly encouraged by your decision to raise our rating. It is our honor to address your concerns, which have been instrumental to our work and will be included in the final version.

---

### Author Response · Authors · 2025-12-03
**Summary of Rebuttal**

Dear Area Chair and Reviewers,

We sincerely thank the Area Chair and all reviewers for their thoughtful evaluations and constructive feedback. We are encouraged that the reviewers recognized the strengths of our work, summarized as follows:

1. **Novel and Clinically Relevant Task** — We introduce *multi-step insight discovery* as a new evaluation paradigm beyond single-turn QA, addressing an important gap in multimodal medical understanding and aligning closely with real clinical needs. (Reviewers 7fob, mEn1, nfBS)

2. **High-Quality, Hierarchical Dataset** — Our expert-validated dataset features a multi-level structure (Goals → Questions → Insights → Evidence) and is carefully balanced in image quality, analytical objectives, and question–insight alignment. (Reviewers eGKb, nfBS, 7fob)

3. **Comprehensive Construction Pipeline** — We provide a full data construction workflow (WSI down-sampling, OCR extraction, LLM-assisted generation, human verification) alongside thorough quality analyses. (Reviewers 7fob, nfBS)

4. **Robust Evaluation Metrics** — Our four complementary metrics (Insight Recall, Precision, F1, Novelty) enable more rigorous and holistic evaluation than prior text-matching methods by jointly capturing quantity and novelty. (Reviewers eGKb, nfBS)

5. **Clear and Reproducible Agent Design** — We propose a modular three-component agent architecture (e.g., VRF–AIA–FQC), provide algorithmic details, and emphasize modularity for transparency and cross-domain adaptability. (Reviewers 7fob, nfBS, eGKb)

6. **Effective Multi-Round Reasoning** — The multi-round chain structure (Root Question → Insight → Follow-up) enhances the depth, diversity, and interpretability of generated insights. (Reviewers eGKb, 7fob)

7. **Strong and Comprehensive Baselines** — We evaluate multiple state-of-the-art LMMs and include a ReAct-style agent as a control, with ablations showing the performance impact of removing key components. (Reviewers 7fob, nfBS)

8. **Extensive Ablation and Qualitative Analyses** — We include ablations, human evaluations, redundancy analyses, and case studies, offering both quantitative and qualitative support for our conclusions. (Reviewers 7fob, nfBS)

---

In addition, we summarize below the key concerns raised during review and our corresponding responses and improvements made during the rebuttal period:

1. **Fairness of Baselines / Potential Bias from Tool Differences** (Reviewers 7fob, mEn1)
    We gave ReAct (GPT-4o) the same tools as our agent. Performance improved but `stayed below MedInsightAgent`. This shows tools help substantially, but our orchestration design `adds extra value beyond just having the right tools`, further validating the effectiveness of our approach.

2. **Reliability of Human Evaluation / Annotator Qualifications / Inter-Annotator Agreement** (Reviewers 7fob, eGKb, mEn1)
   We clarified that all annotators have `medical backgrounds` (medical students) and `added formal agreement metrics` to strengthen the reliability and robustness of our human evaluation.

3. **Does the Novelty Metric Reflect Genuine Clinical Insights or Merely Linguistic Variation?** (Reviewers 7fob, mEn1, nfBS)
   We spot-checked automatic novelty scores with medical annotators. Their `manual ratings aligned closely` with automatic scores, and both showed clear improvement for MedInsightAgent (GPT-4o), supporting the validity and clinical relevance of our novelty metric.

4. **Is the Task Scope Too Narrow?** (Limited to TCGA pathology, no longitudinal data, constrained multimodality) (Reviewers eGKb, nfBS, mEn1)
   We recognize that our current work focuses on TCGA H&E-stained slides as a `specialized-domain benchmark`. We have revised the manuscript to clarify this scope and emphasize its role as a foundational step.

---

Furthermore, we sincerely appreciate the reviewers’ insightful and valuable feedback. In response, we have revised the paper with the following improvements:

1. **Clarified the task definition** to more clearly specify the scope, assumptions, and contributions, reducing ambiguity and improving rigor and reproducibility.

2. **Added a supplementary experiment** comparing ReAct under the same toolset as MedInsightAgent. This addresses fairness concerns and shows that our gains arise from our method’s architectural design rather than tool access.

3. **Incorporated human validation of the novelty metric**, demonstrating strong alignment between automatic and manual scores and increasing the credibility of our novelty assessment.

4. **Added annotator qualifications, detailing their medical background and inter-annotator agreement**. This enhances transparency and strengthens the reliability of our human evaluation.

---

Finally, we hope our detailed responses and new experiments have addressed all concerns, and we sincerely hope our work earns your support.

Sincerely,

Authors of Paper 3899

---

### Meta-Review · Area_Chair_5vvg · 2026-01-10

**Summary:**

This submission describes MedInsightBench, a benchmark for multi-step “insight discovery” in pathology, and a companion MedInsightAgent pipeline. Reviewers agreed that the problem framing is interesting and the paper has a substantial amount of engineering and experimentation. However, the decision is driven by several concerns that affect the validity and interpretability of the benchmark and evaluation.

The most serious issue is a mismatch between inputs and ground truth: the benchmark appears to evaluate models given (goal + image) while many “ground-truth insights” are report-only facts (e.g., HPV/IHC status, margins, node counts) that are not reliably inferable from downsampled H&E images. The authors confirm they do not filter out non-image-inferable insights, positioning this as the challenge the benchmark is meant to expose.

A second issue is the confounding of the agent contribution. The ablation and the added ReAct-with-tools experiment show that specialized tools account for most of the performance gain. The multi-agent orchestration provides only a small incremental improvement over a simpler agent baseline when tool access is matched.

Additional concerns remain around baseline coverage and reproducibility details (comparisons to other medical agent frameworks are not provided; inter-agent communication and evaluation procedures remain under-specified), metric validity (novelty can be inflated by generic statements).

Even after rebuttal additions, the paper does not yet provide a benchmark/evaluation setup where performance differences can be cleanly interpreted as “insight discovery from multimodal medical data.”

**Reviewer Concerns:**

Baselines: Authors added an experiment equipping ReAct with the same tools, showing tool access matters and MedInsightAgent is slightly better than ReAct-with-tools. This helps isolate orchestration somewhat, even if it also reveals the orchestration gain is small.

Annotations: Authors reported inter-annotator agreement statistics (ICC/alpha/correlation) and clarified annotators have medical backgrounds, later specifying they are medical students.

Over-claiming: Authors agreed to scope down “first benchmark” style claims to pathology insight discovery with a goal/question/insight structure. Authors added limited human checks comparing human novelty scores to automatic novelty scores.

Core benchmark: input/label mismatch remains unresolved. The authors confirm models see only (goal + image) and they intentionally keep report-only insights in ground truth without filtering. This leaves a large fraction of evaluation potentially measuring “cannot be inferred from the provided modality,” which undermines interpretability and the “multimodal” positioning.

Agent contribution remains weakly supported. The rebuttal evidence supports the critique that most gains come from tool access; orchestration adds a small margin. The paper’s framing does not reflect this balance clearly.

Ethics/data governance clarity remains insufficient. One reviewer flagged privacy/security/safety due to unclear IRB/de-identification procedures. The rebuttal does not clearly resolve governance for dataset release and handling.

**Reviewer Scores:**

Below is my best estimate of how scores would change with a complete discussion period, based on the tone of the exchange and the rebuttal additions.

Reviewer 7fob states they are generally satisfied and raised score, while requesting remaining issues be addressed in a final version.

Reviewer mEn1 reiterates that major concerns are not solved and argues the tool-matched ReAct result shows limited agent novelty; also remains unconvinced on annotator qualifications and metric consistency.

Reviewer eGKb raised broad concerns (definition of insight, theoretical depth, baseline gaps, rigor, ethics). The rebuttal addresses some points partially but leaves key methodological and ethics transparency issues largely intact; no evidence they were persuaded.

Reviewer nfBS is supportive overall, but with notable limitations; also states “would not mind if paper is rejected,” suggesting they are not strongly anchored above threshold.

---

### Decision · Program_Chairs · 2026-01-26

Reject